

# Quantitative heterodonty in Crocodylia: assessing size and shape across modern and extinct taxa

Domenic C. D'Amore[1], Megan Harmon[1], Stephanie K. Drumheller[2] and Jason J. Testin[3]

[1] Department of Natural Sciences, Daemen College, Amherst, NY, United States of America
[2] Department of Earth and Planetary Sciences, University of Tennessee—Knoxville, Knoxville, TN, United States of America
[3] Department of Physical Science, Physics and Pre-Engineering, Iowa Western Community College, Council Bluffs, IA, United States of America

Corresponding author
Domenic C. D'Amore,
ddamore@daemen.edu

## ABSTRACT

Heterodonty in Crocodylia and closely related taxa has not been defined quantitatively, as the teeth rarely have been measured. This has resulted in a range of qualitative descriptors, with little consensus on the condition of dental morphology in the clade. The purpose of this study is to present a method for the quantification of both size- and shape-heterodonty in members of Crocodylia. Data were collected from dry skeletal and fossil specimens of 34 crown crocodylians and one crocodyliform, resulting in 21 species total. Digital photographs were taken of each tooth and the skull, and the margins of both were converted into landmarks and semilandmarks. We expressed heterodonty through Foote's morphological disparity, and a principal components analysis quantified shape variance. All specimens sampled were heterodont to varying degrees, with the majority of the shape variance represented by a 'caniniform' to 'molariform' transition. Heterodonty varied significantly between positions; size undulated whereas shape was significantly linear from mesial to distal. Size and shape appeared to be primarily decoupled. Skull shape correlated significantly with tooth shape. High size-heterodonty often correlated with relatively large caniniform teeth, reflecting a prioritization of securing prey. Large, highly molariform, distal teeth may be a consequence of high-frequency durophagy combined with prey size. The slender-snouted skull shape correlated with a caniniform arcade with low heterodonty. This was reminiscent of other underwater-feeding tetrapods, as they often focus on small prey that requires minimal processing. Several extinct taxa were very molariform, which was associated with low heterodonty. The terrestrial peirosaurid shared similarities with large modern crocodylian taxa, but may have processed prey differently. Disparity measures can be inflated or deflated if certain teeth are absent from the tooth row, and regression analysis may not best apply to strongly slender-snouted taxa. Nevertheless, when these methods are used in tandem they can give a complete picture of crocodylian heterodonty. Future researchers may apply our proposed method to most crocodylian specimens with an intact enough tooth row regardless of age, species, or rearing conditions, as this will add rigor to many life history studies of the clade.

## INTRODUCTION

What constitutes heterodonty in vertebrates is often difficult to delineate, with different qualitative definitions in place depending on the clade being studied (*Shimada, 2001*). *Kieser et al. (1993*, p.195) referred to the definition of heterodonty as "a bone of contention," and since then the issue has not been fully resolved. Arguably, this lack of clarity is most pronounced within members of Crocodylia. Researchers have often referred to crocodylians as homodont (*Langston, 1973*; *Osborn, 1998*; *Larsson & Sidor, 1999*; *Zahradnicek et al., 2014*). *Peyer (1968*, p.17) defined the term as lacking the discrete dental categories seen in mammals (incisors, canines, premolars, molars), even though he admitted "a sharp distinction between homodont and heterodont is not possible." *Ferguson (1981)* referred to *Alligator mississippiensis* as "pseudoheterodont," because it showed a gradual, as opposed to punctuated, change in tooth shape along the tooth row (see also *Grigg & Gans, 1993*; *Hendrickx, Mateus & Araújo, 2015a*). Size variability along the tooth row has motivated the term "heterometric homodonty" for *Crocodylus niloticus* (*Fruchard, 2012*). Others have applied anisodonty to the clade, which is an apparent change in tooth size but not shape (*Vullo, Allain & Cavin, 2016*). Certain fossil crocodylians, often interpreted as herbivores or omnivores, exhibit multi-cusped and/or grinding teeth, and are specifically called "heterodont" crocodyliforms by researchers (e.g., *Martin, 2007*; *Ősi, Clark & Weishampel, 2007*; *Novas et al., 2009*; *Ősi, 2014*). Lastly, some researchers have argued certain modern crocodylians are actually heterodont, and claim dental categories do in fact exist (*Aoki, 1989*; *Kieser et al., 1993*).

Semantics aside, one reason for the lack of resolution concerning crocodylian heterodonty is that their teeth rarely have been measured. Few studies have performed quantitative shape analyses of crocodylian teeth. Of these, linear-distance measures have been used for fossil identification (*Frey & Monninger, 2010*), replacement rates (*Bennett, 2012*), and biomechanical analyses (*Monfroy, 2017*). Aside from a study evaluating two fossil notosuchians (*Lecuona & Pol, 2008*), and a preliminary geometric morphometric investigation of *Crocodylus niloticus* (*Farrugia, Polly & Njau, 2016*), no studies have quantitatively investigated heterodonty either within or between species. Typically, crocodylian dentition is described qualitatively, with the goal of characterization for phylogenetic analysis, or paleoecological inference (e.g., *Schwarz-Wings, Rees & Lindgren, 2009*; *Young et al., 2012*; *Salas-Gismondi et al., 2015*; *Adams, Noto & Drumheller, 2017*). Qualitative descriptors of crocodylian tooth morphology are numerous, and include terms such as "blunt," "bulbous," "broadened," "button-shaped," "conical," "globular," "fang," "kidney-shaped," "lanceolate," "needle-like," "procumbent," "pseudocanine," "robust," "short," "slender," "spike-like," and "thick" (e.g., *Brazaitis, 1973*; *Groombridge, 1982*; *Aoki, 1989*; *Brochu, 1999*; *Erickson, Lappin & Vliet, 2003*; *Ősi, Clark & Weishampel, 2007*; *Schwarz-Wings, Rees & Lindgren, 2009*; *Fruchard, 2012*; *Gignac & Erickson, 2014*; *Salas-Gismondi et al., 2015*; *Berkovitz & Shellis, 2017*). There is clearly a gap in our knowledge concerning the nature of dental morphology in this clade, and closing this gap may be crucial for a more complete understanding of performance, behavior, and trophic ecology within Crocodylia, as well as more distantly related, crocodylian-line archosaurs.

The lack of quantitative studies on heterodonty in crocodylians and closely related taxa is not due to a lack of applicable methodology, as there has been a burst of morphometric research in non-mammalian teeth in the past decade. Dinosaur teeth have probably received the most attention, with multiple studies using linear-distance measures for the identification of loose fossil crowns or to infer functional paleoecology (*D'Amore, 2009*; *Larson & Currie, 2013*; *Buckley & Currie, 2014*; *Hendrickx & Mateus, 2014*; *Torices, Reichel & Currie, 2014*; *Hendrickx, Mateus & Araújo, 2015b*; *Gerke & Wings, 2016*; *Larson, Brown & Evans, 2016*). Similar measurements have been taken from a number of extinct marine reptiles (*Foffa et al., 2018*). Extant reptiles have been investigated quantitatively as well, including colubrid snakes (*Britt, Clark & Bennett, 2009*; *D'Amore & Juarez, 2018*) and varanid lizards (*D'Amore, 2015*). Prior to this, sharks were studied heavily (*Shimada, 2002*; *Shimada, 2004*; *Shimada & Seigel, 2005*; *Ciampaglio, Wray & Corliss, 2005*). These morphometric analyses have shed light on the nature of heterodonty, dental allometry, and ecomorphology in these vertebrates, and similar methods may be applied to Crocodylia to clarify the state of heterodonty in this taxon.

The purpose of this study is to present a method for the quantification of both size- and shape-heterodonty in members of Crocodylia. Data were collected from a multispecific sample of both extant and extinct specimens housed in museum collections, and their tooth morphology was assessed through two-dimensional geometric morphometric methods. In addition to this major goal, we also (1) outline and describe dental morphology within the specimens sampled; (2) report any morphological consistencies found within the members of our sample; and (3) present the advantages, limitations, and potential future uses of the method. Our intention is to put forward a method for assessing heterodonty that may be applicable to most crocodylian specimens, and may allow for direct comparisons between crocodylians and a variety of other terrestrial and aquatic tetrapods.

## MATERIALS AND METHODS

### Nomenclature

Crocodylian teeth have very few discrete homologous anatomical loci, but, because they exhibit thecodont dentition (sensu *Edmund, 1962*; *Edmund, 1969*), we defined them as having a crown with an apex, a neck, and a root within an alveolus. Nomenclature for tooth morphology used here was proposed by *Smith & Dodson (2003*; Figs. 1A–1B): mesial, towards the point where the premaxillae meet at the midline, or towards the mandibular symphysis; distal, away from the medial premaxillae or the mandibular symphysis; lingual, towards the tongue; labial, towards the ''lips''; basal, towards the base of the tooth or alveolus; apical, away from the alveolus or towards the apex. Crocodylians are known for a condition where a large tooth may be surrounded by much smaller ones. This large tooth is typically referred to as a procumbent tooth (*Gignac & Erickson, 2014*) or a pseudocanine (*Brochu, 1999*), but we simply refer to it as 'enlarged.'

Tooth position was indicated by either the presence of a tooth or an empty alveolus in the host bone (Fig. 1A). Teeth were lettered based on the host bone (premaxilla = P, maxilla = M, dentary = D), and numbered in ascending order from mesial to distal positions

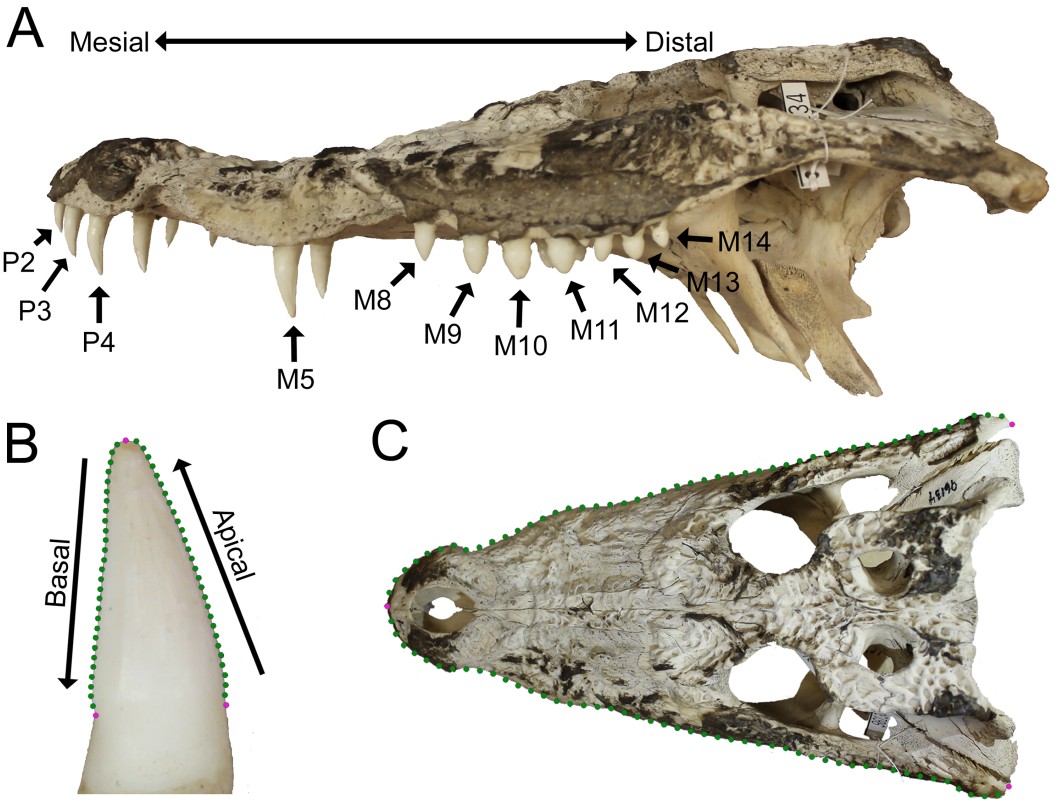

**Figure 1  Data collection methods.** (A) We numbered teeth based on position and host bone (only the left teeth are labeled). Teeth P4, M5, and M10 are defined as the enlarged teeth in this specimen. (B) We photographed each tooth individually, and traced the margins. The software converted each outline into 30 equidistant coordinates. Three coordinates were transformed into landmarks (magenta) and the rest into semilandmarks (green). (C) For skulls, we produced an outline from the dorsal perspective, with 50 coordinates on each margin that were transformed into landmarks (magenta) and semilandmarks (green). (Specimen depicted: *Crocodylus palustris* AMNH 96134).

(the mesial-most dentary tooth was D1, followed by D2, D3, etc.). For consistency, all specimens were assumed to have five premaxillary positions (P1–P5) (*Berkovitz & Shellis, 2017*). Members of *Paleosuchus* and *Osteolaemus* have only four premaxillary teeth during early stages of ontogeny (*Brochu & Storrs, 2012*; *Narváez et al., 2015*), and an alveolus may atrophy in certain species (usually P2) as they grow (*Webb & Messel, 1978*; *Brown et al., 2015*; DC D'Amore, pers. obs., 2017). If only four premaxillary positions were present, position P2 was assumed absent, and skipped over during numbering. In one case, a specimen had six premaxillary teeth (*Alligator mississippiensis*, ROM 4408). This tooth position (P6) was omitted for consistency. Our *Alligator prenasalis* specimen (ROM 1375) had its distal-most cranial positions obscured by poor preservation and matrix. We therefore based its maxillary tooth count on previous osteological accounts (a total of 15 maxillary teeth based on Harvard Museum of Comparative Zoology specimen #1015, (*Mook, 1932*).

## Specimens

Data were collected from 20 species of crown crocodylians, and one species of peirosaurid crocodyliform (Table S1). We included the later as it was readily available with excellent tooth preservation, and allowed us to consider if characteristics indicative of crocodylian heterodonty may be found in species outside the crown clade. The end result was 27 extant and eight extinct specimens, from which we measured 1,263 teeth in total. Although *Caiman crocodilus* is extant, a fossil specimen (UCMP 42844) was also included. Although we did not use any distinct criteria to distinguish juveniles from adults, larger specimens were selected when possible. We chose specimens with the most complete tooth rows in the collections, in that they had the most tooth positions represented by measurable teeth on at least one side of the mouth. Because of normal tooth replacement, post-mortem damage, and/or poor preservation, the percentage of tooth positions represented by measurable teeth was highly variable (Table S2). Modern specimens had tooth rows with 42–100% of their tooth positions represented, and fossil tooth rows ranged 30–78%. Certain fossil specimens only had cranial (*Alligator prenasalis* ROM 1375, *Borealosuchus sternbergii* UCMP 126099, "*Crocodylus*" *affinis* UCMP 131090, *Hamadasuchus rebouli* ROM 52620, *Leidyosuchus canadensis* ROM 1903) or dentary (*Borealosuchus sternbergii* UCMP 131769) material to sample. This variability allowed us to consider in what way the degree of incompleteness along the tooth row may affect the proposed method, as incomplete tooth rows may be common in collections.

## Tooth data collection

Methods were similar to those proposed by *D'Amore (2015)*. We photographed each tooth using either an Olympus Stylus or a Canon Rebel T3 EOS camera with a non-zoom lens. Skulls were positioned on a flat surface with a dark background such as a tabletop or camera stand, and held stationary by an available prop such as a box or sandbag if necessary. A scale was positioned at the same distance from the camera as the tooth. The camera was either mounted on a camera stand, or was held stationary by a researcher (for larger specimens). Digital photographs were taken from the labial perspective (Fig. 1B). For each tooth, we positioned the specimen so the camera lens was parallel to the host bone adjacent to the tooth. This resulted in both carinae being visible in the shot (if present). We simultaneously positioned the lens parallel to the apical-basal long axis, determined qualitatively as when the tooth looked its tallest to the photographer regardless of any labio-lingual curvature. Each tooth was photographed separately, and teeth from both sides were photographed if available. Only fully erupted teeth with the neck visible were included (Fig. 1B). Tooth quality was variable in extant specimens. Teeth with slightly worn apices were included. As the outline of the tooth margin was our basis of measurement, we omitted any teeth with large wear facets or chips that largely interrupted this margin. Cracks down the long axis of the teeth were common, and were omitted if the crack distorted the shape of the tooth or resulted in a space where light could be seen from the other side.

We used a sliding semilandmark analysis (*Bookstein, 1997*; *Sheets, Kim & Mitchell, 2004*; *Zelditch et al., 2004*; *Mitteroecker et al., 2013*) to derive shape measurements from each tooth's outline. Photographs were entered in TpsDig 2.31, and the margin of the tooth was

traced using the curve drawing tool (*Rohlf, 2017a*) (Fig. 1B). Because the enamel margin was not always clear, each tooth was traced from apex to the point where the tooth ceased to taper on the neck for both the mesial and distal side. TpsDig then transformed each of the two traced margins into 30 equidistant coordinates, and we combined the apical-most coordinates. This resulted in three discrete landmarks (two at the base and one at the apex) and 56 semilandmarks (Fig. 1B). This number of coordinates has been used in previous studies of both dinosaur (*Smith, Vann & Dodson, 2005*) and monitor lizard (*D'Amore, 2015*) dentition, as well as claw morphology (*Tinius & Russell, 2017*; *D'Amore et al., 2018*), in which it has been shown to accurately represent the totality of two-dimensional shape (*Tinius & Russell, 2017*). We calculated centroid size (CS), and performed a generalized least squares Procrustes (GLSP) superimposition while sliding the semilandmarks to minimize the total bending energy (*Perez, Bernal & Gonzalez, 2006*; *Gunz & Mitteroecker, 2013*), using the program TpsRelw 1.69 (*Rohlf, 2017b*).

## Skull data collection

The shape of the skull, and particularly the rostrum, has long been considered both an important phylogenetic and ecomorphological feature in crocodylians (*Busbey, 1995*; *Daniel & McHenry, 2001*; *Brochu, 2001*; *Sadleir & Makovicky, 2008*; *Salas-Gismondi et al., 2016*; *Drumheller, Wilberg & Sadleir, 2016*; *Wilberg, 2017*). We attempted to determine if there was a correlation between tooth morphology and head shape, as these traits may be linked. All specimens' skulls were photographed from the dorsal perspective using the same cameras as above (Fig. 1C). Each skull was positioned so the palate was parallel with the tabletop, and the camera was positioned with a camera stand and leveled. A scale was included. We derived skull shape data using a modified version of our technique for tooth outlines. Using TpsDig again, we traced the skull margin from the rostral-most point of contact between the premaxillae to the caudal-most quadratojugal along the margin on each side (Fig. 1C). We chose this margin because it outlined the shape of the head as close to as it would have appeared in life as possible, but avoided internal structures such as the jaw articulations or occipital condyles. Each margin was broken into 50 equidistant coordinates, and the rostral-most coordinates were combined. This resulted in three landmarks (two at the quadratojugals and one at the premaxillary junction) and 97 semilandmarks (Fig. 1C). These also underwent a GLSP superimposition and the semilandmarks were slid to minimize the total bending energy using TpsRelw. In specimens with damaged or missing bones on one side, bilateral symmetry was assumed and the coordinates on the intact side were mirrored. This was achieved by placing another landmark along the mid-sagittal plane at the caudal-most point available. This landmark and the one at the premaxillary junction resulted in a plane that the landmarks were mirrored against.

A body-size metric was needed for several of the following analyses, but unfortunately few were available for all specimens. Commonly used metrics such as snout-vent length and mass were not recorded for most dry skeletons prior to cataloging, and many specimens (especially fossils) lacked femora (see *Farlow et al., 2005*). Therefore, the length of the skull was used as a body size metric (see *Fukuda et al., 2013* for potential influences on this

measurement). We derived skull length from the same landmarks outlining the skull above; it was the linear distance from the rostral-most landmark to the posterior-most landmarks along the mid-sagittal plane (Fig. 1C). (Note: *Borealosuchus sternbergii* UCMP 131769 and *Crocodylus niloticus* AMNH 142494 did not have intact skulls, and were therefore omitted from all analyses involving skull data.)

## Statistical approaches

All analyses were conducted in MorphoJ v. 106d (*Klingenberg, 2011*), SPSS Version 19.0 (IBM Corp, Armonk, NY), and PAST (*Hammer et al., 2001*). If both left and right teeth were available at a given position, we averaged them. For size, CS values were simply averaged together. For shape, each $x$–$y$ coordinate of the GLSP superimposed landmarks and semilandmarks was averaged with its counterpart for both teeth. To ensure that the left and right sides were not significantly different, we ran a 10,000 permutations test on the Procrustes distances between left and right teeth at positions that had both. This test failed to reject the null hypothesis of bilateral symmetry ($p = 0.6785$). If only one tooth was available for a given position, that tooth alone represented said position.

We tested the null hypothesis that there was no statistical difference between tooth rows in our sample. We ran a 10,000 permutations test on the Procrustes distance between cranial (premaxilla and maxilla) and dentary teeth for those that had both.

Skull shape and tooth orientation may be irregularly influenced by captive rearing (*Erickson et al., 2004*; *Drumheller, Wilberg & Sadleir, 2016*), and how this influences tooth shape in crocodylians has yet to be determined. All the modern specimens sampled in this study were categorized as wild caught or 'no data' (Table S2). The latter category indicated the host museum did not know the living conditions of the specimen while alive, and could mean (but does not guarantee) that the specimen was captive raised. To determine if rearing conditions influenced our shape data, we ran a 10,000 permutations test on the Procrustes distances between wild caught and 'no data' specimens in MorphoJ. We excluded *Gavialis gangeticus*, *Mecistops cataphractus*, and *Tomistoma schlegelii*, as they represent an extreme cranial condition and only one individual of each species was available.

A singular measure of heterodonty was derived for each specimen in the form of Foote's morphological disparity [$MD = (\sum_{i=1}^{m} D_i^2)/(m-1)$] (*Foote, 1993*; *Zelditch, Sheets & Fink, 2003*; *Sheets & Zelditch, 2013*). Disparity ($MD$) was the sum of the differences of the values of a given tooth ($i$) from the mean for all teeth from that single specimen ($Di$, also known as the grand mean) squared, with the number of tooth positions ($m$) factored in. We calculated disparity for all occupied tooth positions for each specimen. For size-heterodonty, $Di$ was simply the difference in CS of a tooth from the mean of the specimen (*Zelditch et al., 2004*). For shape-heterodonty $Di$ was the Procrustes distance between the tooth and the mean, and was calculated using DisparityBox7 (*Sheets, 2012*). Heterodonty then was regressed with reduced major axis against skull length to determine if there was a significant allometric change in the clade. To test if the more incomplete tooth rows were strongly affecting heterodonty, we ran both regressions a second time, yet only included specimens that had no less than 70% of their tooth rows represented

by measurable teeth. If the regression statistics were similar to when all specimens were included, this would suggest incompleteness had a minor effect on our results.

We attempted to correlate skull shape to tooth shape between individuals by using a two-block partial least squares (PLS) analysis in MorphoJ. Skull shape represented one block, and average tooth shape represented the other. Average tooth shape was constructed by averaging the corresponding GLSP superimposed landmarks and semilandmarks of every tooth from an individual. The scores for the first PLS of each shape block were plotted against one another and regressed with a reduced major axis. Visualization of variation along each PLS axis was depicted through vector diagrams.

To determine if size and shape were coupled, regression scores of full shape data were generated by MorphoJ (as described in *Drake & Klingenberg, 2008*) and regressed against CS using a reduced major axis. Significance and a high goodness of fit would be indicative of strong coupling between size and shape. A principal component analysis (PCA) was then conducted to visualize the degree of shape variance within all cranial and dentary teeth. PC scores were represented in bivariate plots, and the shape variation of each PC axis was visualized using vector diagrams.

As heterodonty is defined here as variability along the tooth row, we attempted to evaluate the nature of this variability between tooth positions. We used an analysis of variance (ANOVA) to determine if the teeth differed significantly in CS between positions for all specimens sampled. This size metric had unequal variances according to Levene's test, so we specifically ran a Welch's ANOVA. As shape is inherently a multivariate measure, we ran a multivariate analysis of variance (MANOVA) comparing shape between positions. As some specimens had fewer tooth positions than others, the number of teeth occupying the distal-most positions ended up being low. These positions were excluded from the statistical tests, resulting in the only positions considered having at least 7 teeth. To visually represent variability by position, we plotted size and shape against tooth position in a series of box plots. For size, CS was normalized by dividing it by skull length (so as to not obscure the degree of variability in smaller specimens), and then was plotted against tooth position. For shape, PC scores of biologically relevant PCs were plotted against tooth position in a similar fashion.

Preliminary quantitative work has suggested a linear transition in tooth shape along the arcade from mesial to distal (*Farrugia, Polly & Njau, 2016*). We test this by regressing shape data against tooth position using ordinary least squares regressions for each individual. To standardize these regressions, we normalized tooth position into a proportion. We numbered the positions along the tooth row starting with 1 at the mesial-most position, divided each by the total number of positions along the arcade, and then subtracted 0.5 (this subtraction placed the $y$-intercept halfway along the arcade without affecting any ensuing statistics). PCs for each tooth were then regressed against this, and regression statistics were collected. Several factors may be implied by a significantly linear crocodylian tooth row. Slope may be linked to heterodonty, as a steeper slope would imply more shape change along the PC1 scores at the $y$-axis and consequently greater shape-heterodonty. The $y$-intercept represented the shape value at the median position, as the intercept was located half-way along the tooth row. This allowed for direct shape comparison between

all taxa regardless of whether or not the tooth was actually present at said position. To visualize these coefficients, we plotted both slope and $y$-intercept for each regression in scatterplots for both the cranium and the dentary.

## RESULTS

### Shape variability in the sample

No statistically significant difference was found between cranial and dentary tooth rows ($p = 0.2455$) or wild caught versus 'no data' modern specimens ($p = 0.4229$), failing to reject the null hypothesis under both circumstances. When regression scores were plotted against CS, the regression had a goodness of fit accounting for less than 10% of the variance ($y = 0.313x\text{-}0.981$; $r^2 = 0.090$; $p < 0.0001$; $95\% = 0.296, 0.328$).

Most of the shape variance in Crocodylia was along a single PC axis. PC1 accounted for over 92.11% of the variance (Table S3). The shape changes towards negative PC1 scores include apical-basal elongation, narrowing at the base, and a gentle concavity on the distal margin (Fig. 2A). For simplicity, we will refer to this extreme as 'caniniform' (*Erickson et al., 2012*; *Erickson et al., 2014*; *Gignac & Erickson, 2014*). Positive shape changes depicted an apical-basal shortening and mesial-distal broadening, and we will refer to the extreme as 'molariform' (*Erickson et al., 2012*; *Erickson et al., 2014*; *Gignac & Erickson, 2014*). We describe PC2 (3.22% of the variance) as the orientation of the tooth, or how much it 'leans' (Fig. 2A). Positive values indicate the apex leaning in the mesial direction, and negative values indicate a lean in the distal direction. Note that this is not a measure of curvature, as neither margin changes its concavity or convexity. We do not consider any other PCs, as the amount of variance represented by them is very low (Table S3).

Figure 2B illustrates the morphospace produced by PC1 and PC2 scores. At the superfamily level, there was a large amount of overlap between alligatoroids and crocodyloids. Most specimens had teeth ranging a large portion of the PC1 score spectrum, with little separation between them. The only exceptions were some crocodyloid teeth below PC1 scores of $-0.2$. Both *Borealosuchus sternbergii* tooth rows had PC1 scores between $-0.22$ and $0.20$, and *Hamadasuchus rebouli* ranged between $-0.19$ and $0.23$. These were nested within the alligatoroids and crocodyloids. *Gavialis gangeticus* deviated from the rest in that most teeth had PC1 scores of $<-0.20$ (Fig. 2B). PC2 scores did not differ between groupings of taxa, as each group occupied most of the shape range.

### Skull vs. tooth morphology

Skull and average tooth shape were significantly correlated. For the two-block PLS test, PLS1 encompassed 99.96% of total covariation and had a correlation coefficient of 0.7937. Shape variability within the skull shape block showed the snout transitioning from narrow to broad (Fig. 3A). Taxa that occurred below a PLS1 score of $-0.15$ were the slender-snouted taxa as defined by *Brochu (2001)*, including *Gavialis gangeticus*, *Mecistops cataphractus*, and *Tomistoma schlegelii*. The remainder of the species, defined as either generalized or blunt-snouted (also by *Brochu, 2001*), occurred around the mean and positive half mixed together. Shape variability within the tooth shape block was similar to the above PCA of tooth shape, displaying a transition from caniniform to molariform with increasing values
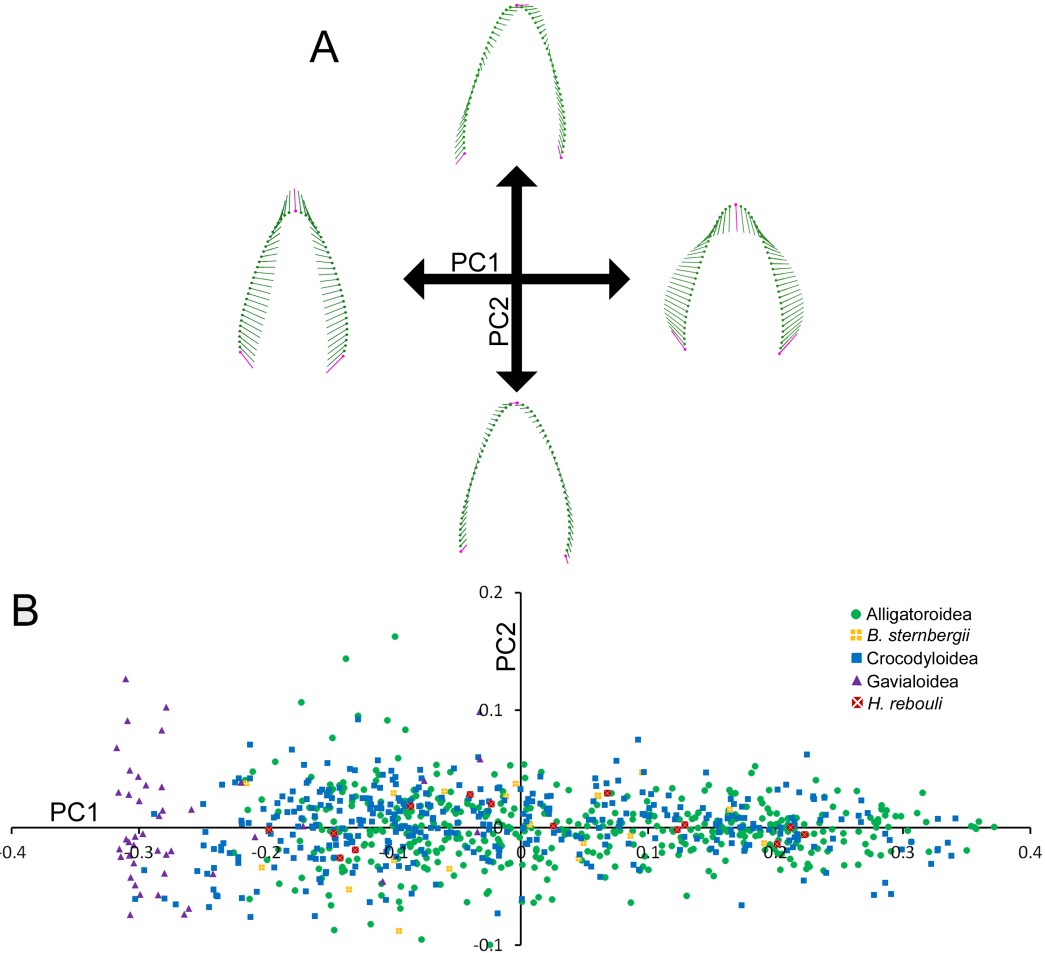

**Figure 2** **Variability within the first principal component for teeth.** (A) Vector diagrams indicate the maximum range of variance (vectors) from the mean (points) for both cranial and dentary teeth. Landmarks are in magenta and semilandmarks are in green. (B) Scores from the first and second principal components are plotted in a morphospace, with major taxonomic groupings labeled.

(Fig. 3A). These blocks regressed significantly against one another (Fig. 3B), with slender-snouted taxa separating out with the most caniniform teeth. Scatter increased around the means, indicating the correlation was not as strong among the generalized-to-blunt snouted taxa. *Alligator prenasalis*, *Brachychampsa* sp., and "*Crocodylus*" *affinis* were all relatively blunt-snouted, but rose noticeably above the regression. This suggested they possessed much more molariform teeth on average than their counterparts of similar skull shape.

## Foote's disparity and heterodonty

Size-heterodonty was significantly correlated with skull length, with an $r^2$ of 0.760. The largest individuals according to skull length (members of *Crocodylus niloticus* and *Crocodylus porosus*) possessed the greatest unadjusted size-heterodonty (Fig. 4A). Members of *Alligator* had negative residual size heterodonty, with *Alligator prenasalis* as the lowest. On the other side of the regression, residuals of caimanine specimens (*Caiman*, *Paleosuchus*)
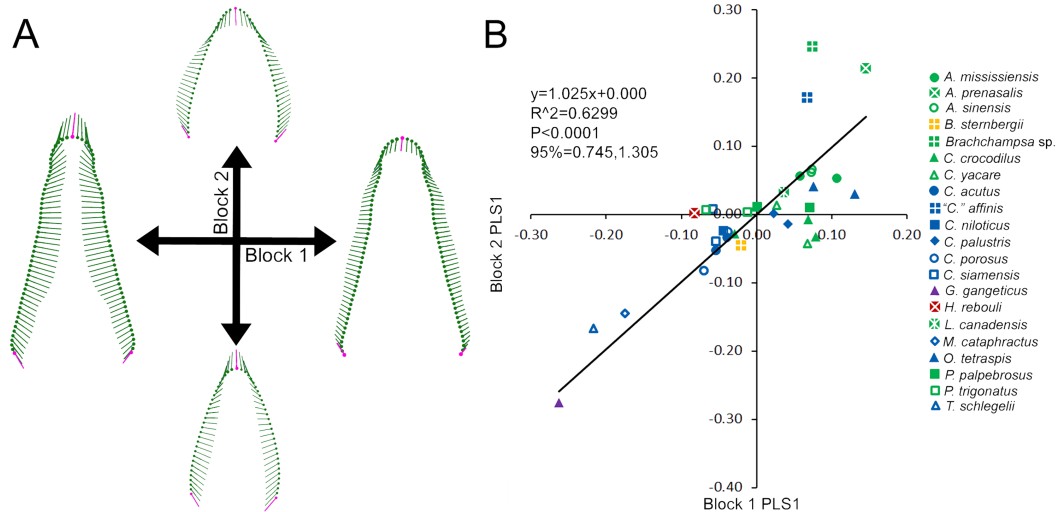

**Figure 3** **Partial Least Squared (PLS) two-block analysis of shape.** (A) Vector diagrams indicate shape variance of PLS1 for skull shape (Block 1) and average tooth shape (Block 2). (B) PLS1 scores for both blocks were regressed, with colors and shapes representing species. Regression information is listed.

were all positive with the exception of one individual. Several members of *Crocodylus* had values around or below zero, but one *Crocodylus porosus* specimen had a high residual. One *Osteolaemus tetraspis* individual had the highest size-heterodonty residual, with the other close to the regression. *Hamadasuchus rebouli* had positive residuals, similar to the larger *Crocodylus porosus* and the caimanine specimens. The slender-snouted taxa (*Gavialis gangeticus*, *Mecistops cataphractus*, *Tomistoma schlegelii*) had some of the more negative residuals. For the regression excluding individuals with less than 70% of either tooth row represented, the regression statistics ($y = 2.208x - 8.583$; $r^2 = 0.770$; $p < 0.0001$) were strikingly similar to those of the complete dataset ($y = 2.196x - 8.549$; $r^2 = 0.760$; $p < 0.0001$).

Shape-heterodonty was highly variable. Shape regressed significantly, but had a rather poor correlation ($r^2 = 0.157$), with skull length (Fig. 4B). Similar to size-heterodonty, the slender-snouted taxa had some of the lowest shape-heterodonty, although *Tomistoma schlegelii* was greater than several other taxa. In addition, members of *Brachychampsa* sp. and *Alligator prenasalis* also had some of the lowest shape-heterodonty in our sample. *Crocodylus siamensis* specimens were more shape-heterodont than their congenerics, with one individual being the most shape-heterodont in our sample. Several caimanine individuals, and both *Osteolaemus tetraspis* specimens, also had relatively high shape heterodonty. For the regression excluding individuals with less than 70% of either tooth row represented, regression statistics ($y = -0.648x - 0.235$; $r^2 = 0.238$; $p < 0.0397$) were also similar to those of the complete dataset ($y = -0.715x - 0.172$; $r^2 = 0.157$; $p < 0.0223$).

## Heterodonty along the tooth row

Tooth position count varied between species (see Table S2). Most alligatoroids had between 19–20 positions on the cranial tooth row. Many had a similar number on the

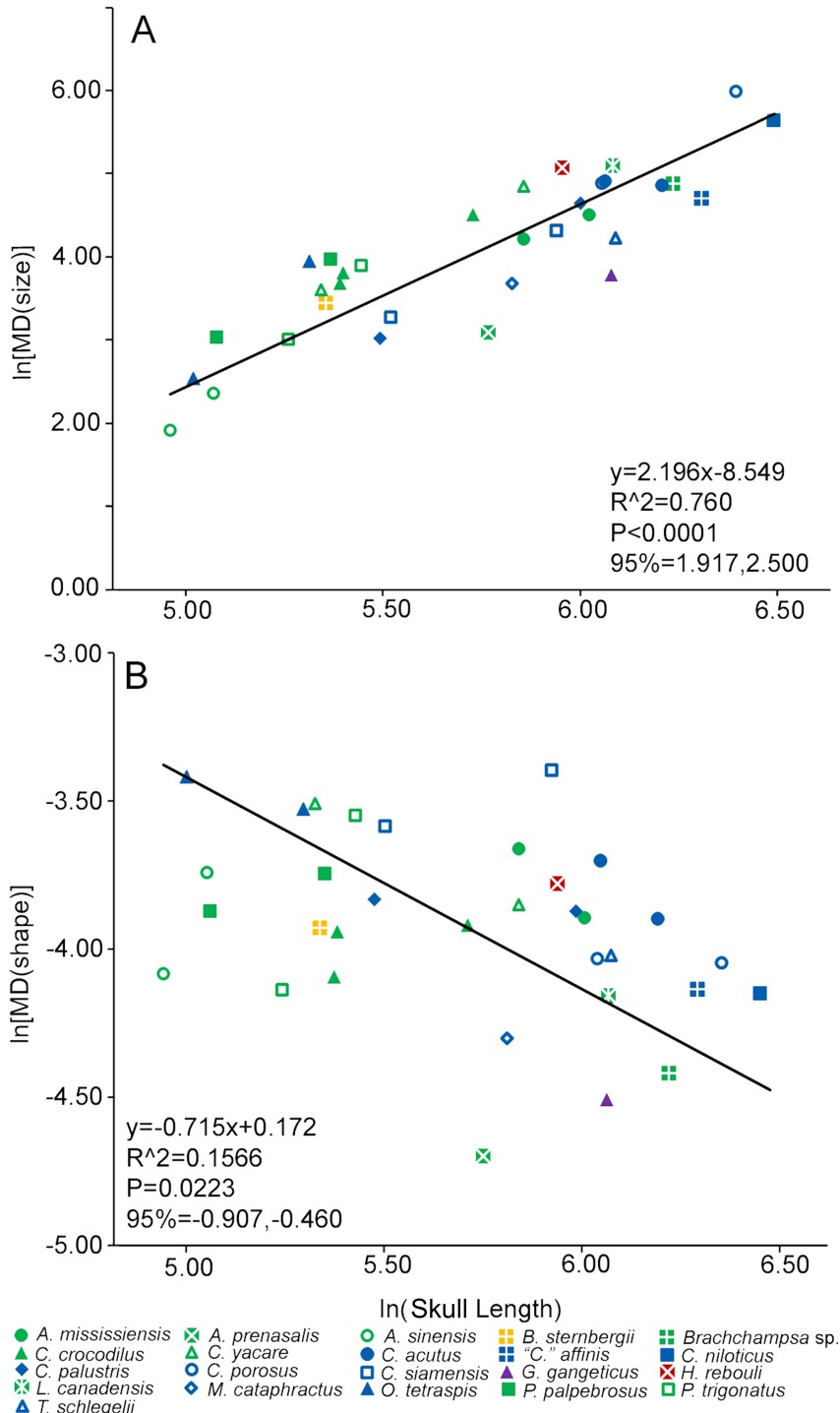

**Figure 4  Heterodonty represented by Foote's morphological disparity.** Ln scaling of Morphological Disparity (MD) for size (A) and shape (B) are plotted against the ln of skull length. Colors and shapes represent species, and regression information is listed.

dentary, except that members of *Paleosuchus* had 22 positions. *Leidyosuchus canadensis* had the most (23) cranial positions of the alligatoroids. *Hamadasuchus reboulii* (20 cranial) and *Borealosuchus sternbergii* (23 cranial and 20 dentary) specimens fit within ranges of alligatoroids. Members of *Crocodylus* and *Mecistops cataphractus* had between 18–19 cranial and 15 dentary positions. *Osteolaemus tetraspis* specimens had the least tooth positions for any crocodyloid (17 cranial and 14 dentary), and *Tomistoma schlegelii* had the most (21 cranial and 19 dentary). The *Gavialis gangeticus* specimen had more positions than any other species sampled (28 cranial and 26 dentary). These tooth counts are similar to previous published accounts (*Brown et al., 2015*; *Berkovitz & Shellis, 2017*).

In all taxa combined, ANOVA indicated size differed significantly between tooth positions [$F(34, 261.45) = 4.57$; $p < 0.0001$]. In Alligatoroidea and Crocodyloidea, size undulated three times along the dental arcade resulting in significant differences between positions for both the cranium and mandible (Figs. 5A–5B). Each undulation peaked with an enlarged tooth. These were typically represented by P4 for both clades, and M4 for alligatoroid and M5 for crocodyloid specimens (sensu *Brochu & Storrs, 2012*). In addition, members of *Paleosuchus* had very large P3 and M3. "*Crocodylus*" *affinis* also had a large P3. Unlike other alligatoroid specimens, *Leidyosuchus canadensis* had both M4 and M5 enlarged, and the *Brachychampsa* sp. had an enlarged M5 like crocodyloids (sensu *Norell, Clark & Hutchison, 1994*). A final undulation resulted in an enlarged tooth at M9-11 (Figs. 5A–5B). Interspersed between these were smaller teeth, with the distal-most tooth often the smallest. The dentary was similar to the cranium, with three undulations in size. Enlarged teeth were found at positions D1 and D4, with a third size-peak between D11 and D14. Note that the position of the enlarged teeth along the cranial tooth row tended to align with smaller teeth along the dentary tooth row, and vice versa. This resulted in an 'adjoining' pattern between the size peaks of one arcade and the valleys of the other. The gavialoid specimen differed markedly by having the two mesial-most teeth enlarged, and the remainder showed a gradual decease in size distally (Fig. 5C). *Hamadasuchus rebouli* had some of the largest teeth for its skull length with a dramatic variation in size.

Shape also differed significantly between positions according to MANOVA [$F(4046, 23939.54) = 1.27$; $p < 0.0001$; Wilk's $\Lambda = 0.002$; partial $\eta^2 = 0.171$], although in a visibly different fashion than size. We only graphed PC1 scores against tooth position, as the other PCs represented under 5% of the variance each and were not considered biologically relevant to tooth position (for a justification, see Supplemental Information 1). Alligatoroids and crocodyloids both had mesial teeth that were typically the most caniniform in the mouth, and distal teeth the most molariform (Figs. 5D–5E). In cranial teeth P1-M4 tended to exist primarily between PC1 scores of $-0.25$ and $0.00$, followed by a gradual increase in score values as positions became more distal. Dentary teeth represented a more uniformly gradual caniniform-to-molariform transition. Both superfamilies were highly variable. Alligatoroidea had teeth generally more molariform, with upper outliers almost entirely represented by *Alligator prenasalis*, *Alligator sinensis*, and *Brachychampsa* sp. (Fig. 5D). Crocodyloidea was generally more caniniform, with mesial upper outliers represented by "*Crocodylus*" *affinis* and lower outliers represented primarily by *Tomistoma schlegelii* (Fig. 5E). Both *Borealosuchus sternbergii* tooth rows had PC1 scores between

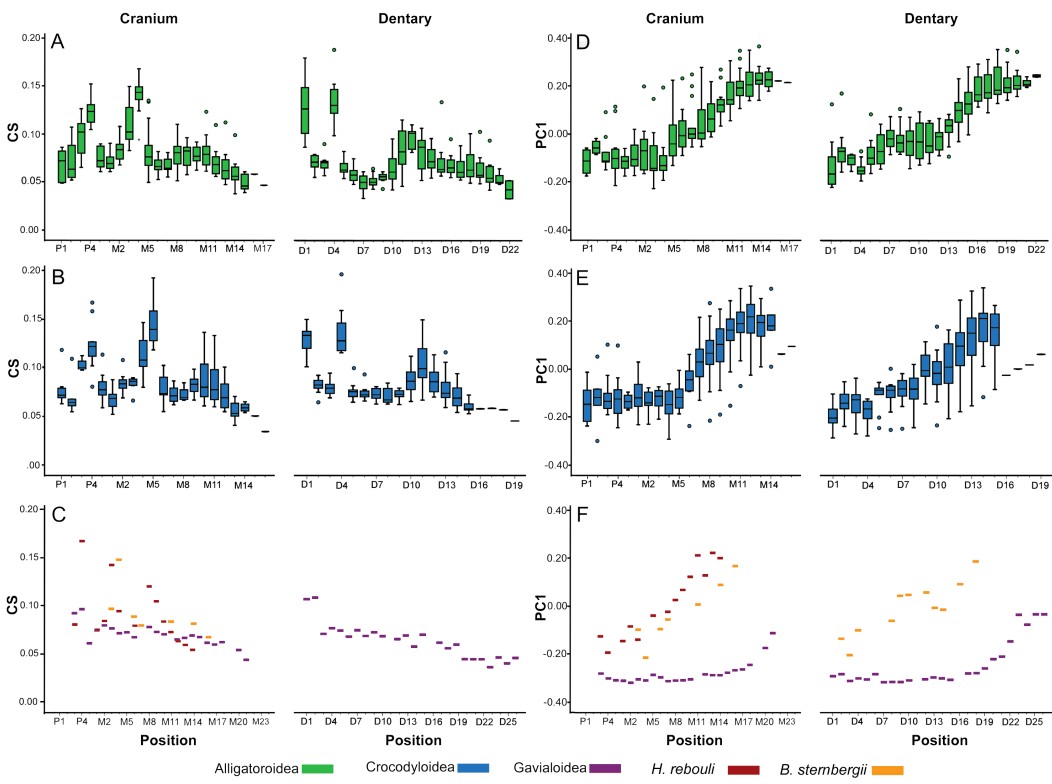

**Figure 5  Heterodonty by tooth position.** Centroid Size (CS) and principal component one (PC1) for extant Alligatoroidea (A, D), Crocodyloidea (B, E), and remaining taxa (C, F), plotted against position along the arcade. Colors represent major taxonomic groups. See Fig. 2 for a visual representation of shape change depicted by PC1 scores.

−0.22 and 0.20, and *Hamadasuchus* ranged between −0.19 and 0.23 (Fig. 5F). Both taxa showed a similar progression from caniniformy to molariformy as the alligatoroids and crocodyloids. *Gavialis gangeticus* deviated from the others the most, where most teeth had scores of <−0.20 with a steep increase towards the average in the distal-most fifth of the arcade (Fig. 5F).

Tooth shape was strongly influenced by tooth position (Fig. 6). When each modern individual's PC1 values were regressed against position, all linear regressions were significant (Table S2). The vast majority of tooth row regressions had $r^2$ values above 80%. *Gavialis gangeticus* had the lowest $r^2$ values (cranium = 0.495, mandible = 0.616), followed by the crania of the fossil *Caiman crocodilus* ($r^2 = 0.728$) and *Tomistoma schlegelii* ($r^2 = 0.747$). Both cranial and dentary tooth rows typically had slopes between 0.25–0.55 (Fig. 6). More shape heterodont taxa typically had greater slopes, with *Crocodylus siamensis* specimens having some of the steepest slopes (0.39–0.68). All the slender-snouted specimens had the *y*-intercepts between −0.25 and −0.14, indicating strong caniniformy at the median position. Living members of *Alligator* had *y*-intercepts between 0.029 and 0.085, indicating molariformy at the median (Fig. 6). *Alligator prenasalis, Brachychampsa* sp., and "*Crocodilus*" *affinis* had the shallowest slopes (0.22–0.32) and the greatest *y*-intercepts

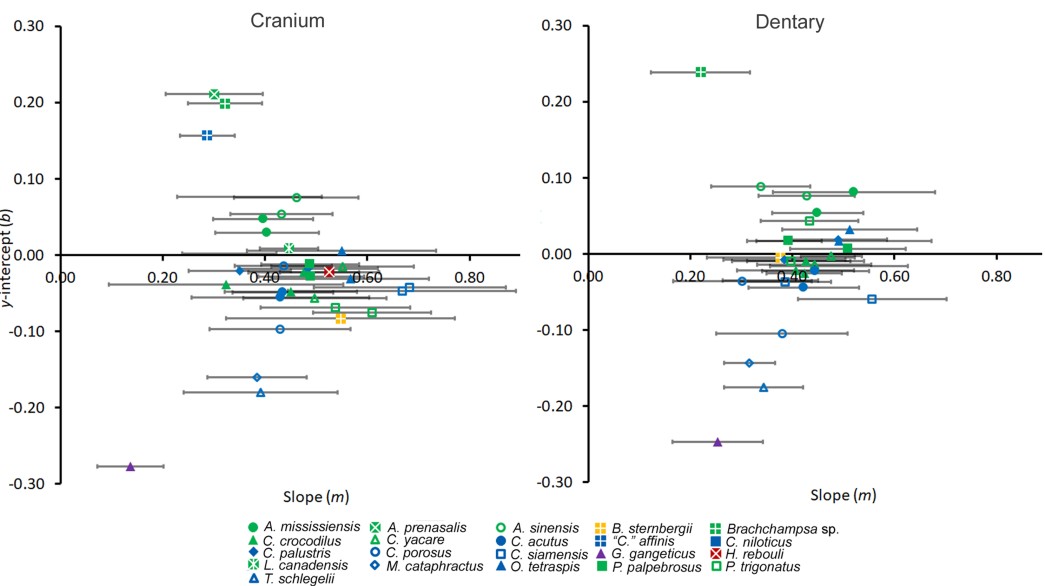

**Figure 6  Regression information for shape heterodonty.** Slope (*m*) and y-intercept (*b*) data for regressions of the first principal component plotted against tooth position for individuals. Error bars indicate 95% confidence intervals, and colors and shapes represent species. Regression statistics are available in Table S2.

(0.15–0.24) in our sample, indicating molariform teeth are consistent along the tooth row. *Hamadasuchus rebouli*'s regression characteristics are similar to members of *Caiman* and *Osteolaemus tetraspis*. The slope of the fossil *Caiman crocodilus* differed from modern members of *Caiman* by being much shallower (Fig. 6).

## DISCUSSION

### Defining heterodonty within Crocodylia

The methods proposed here offer a multi-faceted approach to quantifying heterodonty in Crocodylia. As was first proposed in *D'Amore (2015)*, outlining the margin of the tooth is a comprehensive method for measuring two-dimensional tooth shape. This type of semilandmark analysis is ideal for any cylindrical, conical, caniniform, or ziphodont dentition, and would include many archosaurs, squamates, sauropterygians, ichthyosaurs, teleosts, sharks, etc. As tooth morphology is often compared between taxonomically disparate groups (examples in *Ciampaglio, Wray & Corliss, 2005*; *Foffa et al., 2018*), future studies should use semilandmark analyses to compare crocodylians to these groups to answer numerous phylogenetic and ecomorphological questions.

Foote's morphological disparity allows for a continuous and quantitative measure of heterodonty that may be used for comparison between taxa, or compared with other variables (as was done here with skull length). This measure is ideal if one is interested in *how much* heterodonty is apparent. Alternatively, if one is interested in what shape characteristics make up tooth heterodonty, ordination approaches suffice in describing shape variability. Although plotting PC scores as Cartesian coordinates in a morphospace
is traditionally done to visualize shape variability (as in Fig. 2B), overlap due to shape-heterodonty makes specimens, species, and even superfamilies almost indistinguishable from one another. This exemplifies why methods such as box plots and linear regressions may be preferable over the more 'standard' morphospace depiction when heterodonty is concerned. It is convenient that only PC1 accounted for over 90% of the variance, allowing for us to use it as the sole measure of shape here. If more PCs accounted for over 5% of the shape variance, it would be appropriate to depict those other PCs in the same manner as PC1 for a comprehensive understanding of shape.

Regression analysis of shape against position generally yielded strong correlations (Table S2), and the resulting coefficients were useful for comparison between individuals. Certain caveats should be considered though. Regression may be more appropriate for the dentary than the cranium, because in the cranial arcade the mesial-most teeth do not appear to differ from one another as much as the remainder. In most crocodylians this effect is mild, and $r^2$ values are still high. The effect is very pronounced in *Gavialis gangeticus* though, as both the premaxillary and most maxillary teeth are similar in shape. It is also interesting that this happens in the dentary as well. Future studies should consider this when applying this method to Gavialoidea or dentally analogous taxa.

Although the task of assigning a singular dental morphotype to any one species of crocodylian is beyond the scope of the study, our data suggest that it would be potentially difficult. Heterodonty seems to vary within species, making the assignment of a singular heterodonty measure to an entire species dubious. As far as biological explanations for this, tooth form is almost certainly influenced by allometry. Ontogenetic shifts in feeding niche have been documented in modern crocodylian species (e.g., *Groombridge, 1982*; *Webb, Manolis & Buckworth, 1982*; *Pooley & Gans, 1976*; *Pooley, 1989*; *Delany, 1990*; *Santos et al., 1996*; *Da Silveira & Magnusson, 1999*; *Subalusky, Fitzgerald & Smith, 2009*; *Wallace & Leslie, 2008*; *Borteiro et al., 2009*; *Hanson et al., 2014*), and allometric changes in the feeding apparatus with size are often explained as a structural consequence of this (e.g., *Dodson, 1975*; *Webb & Messel, 1978*; *Hutton, 1987*; *Erickson, Lappin & Vliet, 2003*; *Verdade, 2000*; *Wu et al., 2006*; *Watanabe & Slice, 2014*; *Gignac & Erickson, 2016*; *Gignac & O'Brien, 2016*). Concerning teeth, a qualitative increase in overall molariformy was observed in *Alligator mississippiensis*, as it functioned to meet the mechanical demands of increased durophagy (*Erickson, Lappin & Vliet, 2003*; *Gignac & Erickson, 2014*). Although our sample size is too low to confidently assess dental ontogeny within each species, we did see a similar general trend in conspecifics of different sizes. In particular, the larger of our two *Crocodylus porosus* had a greater *y*-intercept indicating greater molariformy.

Although there was no significant effect of captive rearing on our specimens, we believe this principle should still be investigated further. Some cases of captive rearing have resulted in very different, and easily distinguishable, cranial morphologies, but there is also quite a bit of overlap concerning others (*Drumheller, Wilberg & Sadleir, 2016*). This may, or may not, happen with teeth. A larger data set, looking at multiple ontogentic stages, is necessary to determine if rearing condition has any measurable effects on teeth. The method proposed here would allow for the rigorous comparison between these two rearing conditions. It

would determine if any changes do occur, as well as other factors may correlate to said changes.

## Morphological trends within Crocodylia

All crocodylian specimens measured here were heterodont to varying degrees, and these data showed significant variability of morphotypes along the dental arcade for all specimens (Fig. 7). Although dentition varied between species, certain consistencies were seen throughout the clade:

1. Similar teeth occurred on both the cranial and dentary dental arcades.
2. As body size increased, size-heterodonty increased reliably with it. Shape-heterodonty shows a much less reliable negative correlation with body size.
3. The vast majority of shape variance from the labial perspective occurred along a single shape axis, representing the transition from caniniform to molariform.
4. There was serial homology in tooth shape from-mesial-to-distal along the tooth row, and molariformy increased in this direction. The transition was significantly linear for both dental arcades in all specimens.
5. Size variability consisted of a non-linear, undulating pattern with three peaks that adjoin with the opposing row, with enlarged crowns interspersed within smaller crowns. This corresponded with the festooning pattern seen in the tooth bearing bones, and was less apparent in slender-snouted taxa.

Size- and shape-heterodonty were very loosely coupled in Crocodylia as they changed in dramatically different, and primarily independent, fashions along the arcade. Some correlation did occur; the regression's significance was probably the result of the fact that the distal-most crowns were typically both the smallest crowns as well as the most molariform. Nevertheless, the undulating pattern of tooth size did not align with linear shape heterodonty for the vast majority of the tooth row, as indicated by the very low $r^2$.

The low degree of coupling begs the question; do developmental agents influence size and shape separately? Although quite a bit of research has looked at how crocodylian teeth grow and replace themselves (*Edmund, 1962*; *Westergaard & Ferguson, 1986*; *Westergaard & Ferguson, 1987*; *Westergaard & Ferguson, 1990*; *LeBlanc et al., 2017*), surprisingly little has been done on what developmental influences affect tooth size and shape. Modern crocodylians replace their teeth in waves, or Zahnreihe (*Edmund, 1962*; *Westergaard & Ferguson, 1990*; *Osborn, 1998*), but it is unclear how the nature of these waves relate to the morphological variables investigated here. *Kieser et al. (1993)* compartmentalized the dentition along the tooth row for *Crocodylus niloticus*, grouping teeth into 'incisor,' 'premolar,' and 'molar' regions. These designations attempted to account for both size and shape heterodonty; each was defined by an enlarged tooth, and each become progressively more molariform. They did not offer a developmental mechanism that differentiated these categories though. *Fruchard (2012*, p.7) suggested that the only difference between enlarged teeth and their smaller counterparts was that the former was "programmed to be bigger," suggesting some sort of additional developmental signaling to enlarge teeth. More research is needed on how tooth shape and size are established developmentally in order to truly understand what generates heterodonty.

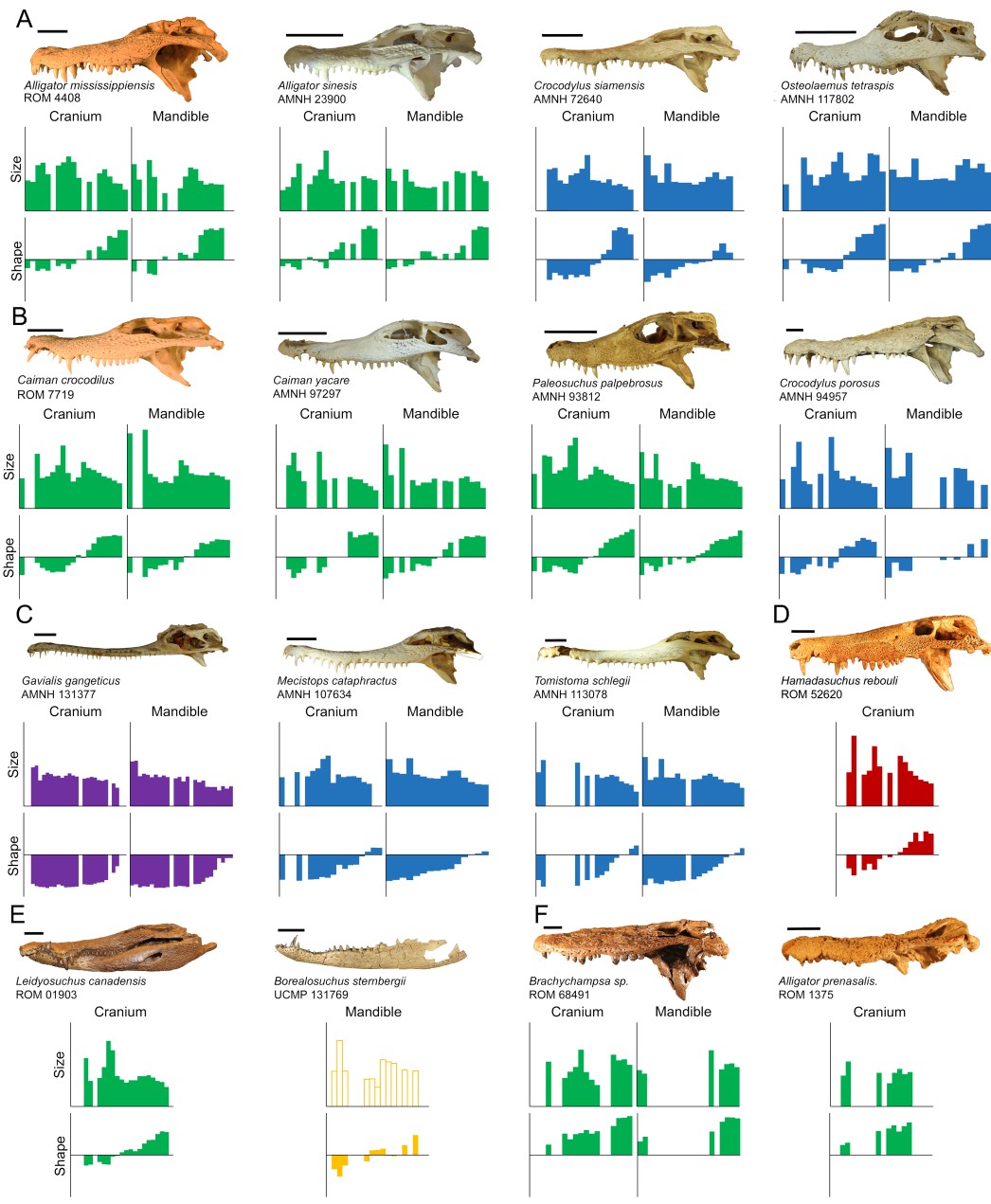

**Figure 7  Direct comparisons between selected extant and extinct taxa.** The size axis represents normalized centroid size (ranging from 0.00 to 0.20), and the shape axis represents scores from the first principal component (ranging from −0.04 to 0.04). Taxa are grouped by (A) modern specimens with high molariform distal teeth, (B) modern specimens that display high size heterodonty, (C) slender-snouted specimens, (D) *Hamadasuchus rebouli*, (E) *Borealosuchus sternbergii* and *Leidyosuchus canadensis*, and (F) fossil globidont specimens. (Note: *Borealosuchus sternbergii* teeth were not size normalized by its own skull length, as indicated by hollow bars). Scale = 5 cm.

The fact that *Hamadasuchus rebouli* showed similar trends in heterodonty to crown crocodylians was particularly revealing. This indicated the methods proposed here may be transferable to at least some crocodyliforms outside of Crocodylia. It should also be noted that this consistency exists in spite of the numerous craniodental characteristics that differ between *Hamadasuchus rebouli* and the crocodylians, including a deep-snout and ziphodont teeth (*Larsson & Sues, 2007*). This characteristic heterodonty may be covergently derived or homologous with what is seen in this peirosaurid, and further research should investigate how frequent it is seen in Crocodyliformes as a whole.

## Adaptive explanations for morphological variability in modern taxa

There is a wide range of tooth morphologies present in modern Crocodylia, and, as teeth are anatomical units used for feeding and aggression, functional inferences may be drawn based on our present understanding of behavior and performance. Bite force in crocodylians is primarily influenced by size (*Erickson et al., 2012*; *Erickson et al., 2014*), and our data set shows that similarly sized crocodylians may have very different tooth dimensions. This rules out adductor-generated maximum bite force as the sole limiting factor dictating tooth form. Although we are reluctant to associate specific prey items with specific tooth forms, size and shape will influence how a tooth interacts with food items possessing certain physical properties. We therefore suggest that a biomechanical link should exist between the structural limits imposed by tooth form and the material properties of the substrates with which it interacts.

As with all jawed vertebrates, crocodylian teeth will succumb to different speeds and pressures based on their respective position along the arcade. Caniniform mesial teeth are ideal for the initial acquisition of prey. Pointed apices reduce surface area to puncture compliant foods that deform under pressure, such as muscle, fat, and fibrous connective tissue (*Frazzetta, 1988*). Being farther from the hinge, these teeth move faster during a strike and are more likely to contact prey trying to escape (*Busbey, 1989*). They will also endure less pressure during a bite based on their position (*Erickson et al., 2012*), and can afford to be relatively elongate and gracile. On the other end, distal teeth need to withstand greater tooth pressures due to their close proximity to the hinge (*Cleuren, Aerts & Vree, 1995*; *Erickson, Lappin & Vliet, 2003*; *McHenry et al., 2006*; *Erickson et al., 2012*). This explains why these teeth are typically on the molariform half of the shape spectrum; the larger base-to-height ratio gives them greater relative bending strengths to withstand said pressures (*Van Valkenburgh & Ruff, 1987*; *Gignac & Erickson, 2014*; *Monfroy, 2017*). Because force is highest in this region, it is ideal for processing food items after they are acquired (*Busbey, 1989*; *Davenport et al., 1990*; *Cleuren & De Vree, 2000*). The reduced height of these teeth also ensures they do not impede jaw closure. This necessity is very apparent in our representative member of *Gavialis gangeticus*, and provides a functional explanation for the poor linear shape relationship along the tooth row in this individual. Having all the teeth be highly caniniform except for the distal-most region may be interpreted as an attempt to reduce heterodonty as much as possible (*Grigg & Gans, 1993*), while ensuring the distal crowns do not impede jaw closure or break when processing food.

Particular attention should be paid to the relative size of the distal-most crowns, as they vary considerably within our sample. Most modern alligatoroids and crocodyloids have a single enlarged tooth followed distally by several smaller teeth on the caudal half of their jaw. These teeth typically had positive PC1 scores, especially within *Alligator mississippiensis* and *Crocodylus siamensis*, and were also some of the smallest teeth in their arcades (Fig. 7A). Members of *Alligator sinensis* differed from this though, in that they had a row of 4–5 relatively large, high-molariform crowns (followed by only one crown reduced in size). Probably the most extreme condition, *Osteolaemus tetraspis* specimens had distal crowns that were exceptionally large; the largest relative crowns at positions M10-12 and D11-13 for modern taxa all belonged to members of this species. These two species also have the lowest number of teeth for modern alligatoroids and crocodyloids respectively, a reduction potentially based on the need to fit these enlarged teeth. *Aoki (1989)* qualitatively noted these unique conditions, and suggested they facilitated durophagy. All alligatoroids and crocodyloids sampled here have been recorded to consume at least some hard prey items though (e.g., *Brazaitis, 1973*; *McIlhenny, 1976*; *Taylor, 1979*; *Groombridge, 1982*; *Ross & Magnusson, 1989*; *Santos et al., 1996*; *Selvaraj, 2012*; *Nifong & Silliman, 2013*), so it is unclear what selection pressure resulted in these particular morphologies. It may be a result of body size. Bite force tests of *Alligator mississippiensis* showed the pressure produced at its enlarged M11 to be adequate to crush its harder prey items (*Erickson, Lappin & Vliet, 2003*; *Gignac & Erickson, 2014*). If this is the case in most of the large crocodylians, enlarging the distal-most crowns would be unnecessary. *Alligator sinensis* and *Osteolaemus tetraspis*, on the other hand, may need more extreme dentition closer to the hinge; their smaller size would make it more difficult to process foods with similar mechanical properties. Another explanation for this may be the frequency of consuming hard prey. Although both these species have broad diets, studies have shown certain (but not all) populations of these species to consume disproportionately large numbers of shelled mollusks and crustaceans (*Groombridge, 1982*; *Ross & Magnusson, 1989*; *Luiselli, Akani & Capizzi, 1999*; *Pauwels et al., 2007*).

All taxa measured here also have two sets of enlarged mesial teeth on both arcades. These teeth are well built for puncturing, likely to make first contact with prey during jaw closure, and resilient against struggling prey (*Iordansky, 1964*). An apparent trade-off to enlarging these teeth is the need to reduce the size of teeth on the opposing tooth row. This character played a large role in size-heterodonty, with different crocodylians undulating their tooth sizes to different degrees. High relative size-heterodonty in specimens found within Caimaninae was typically a consequence of the dramatic size difference between the enlarged teeth and the small remaining crowns, (Fig. 7B). Their dentary crowns in particular became so large they often grew entirely through the cranial rostrum in adults (as mentioned in *Brazaitis, 1973*), which suggests securing prey takes priority. The remaining crowns were rather small by comparison, including the distal crowns: the teeth with the greatest mechanical advantage when processing hard prey. This overall condition may be specialized for hunting more mobile and/or compliant prey, as these types of prey may be punctured quickly and securely with the enlarged, pointed crowns (*Sampaio et al., 2013*). The *Crocodylus porosus* specimens had the largest M5 crowns in our sample, which may

also show a prioritization for puncturing and securing soft-bodied prey in a larger context (Fig. 7B). This species is notorious for actively hunting large vertebrates such as sharks, cattle, horses, and humans (e.g., *Taylor, 1979*; *Kar & Bustard, 1983*; *Groombridge, 1982*; *Doody, 2009*; *Hanson et al., 2014*), and these teeth are ideal for puncturing and securing such prey. Similar to the caimanine specimens, this species atrophies position P2 to make room for its enlarged D1 crowns (*Brown et al., 2015*).

The slender-snouted species possessed generally more caniniform teeth, which may be a consequence of feeding habitat and prey preference. These taxa have a reputation for eating small, aquatic prey with a focus on fish (*Peyer, 1968*; *Webb, Manolis & Buckworth, 1982*; *Erickson et al., 2012*), and multiple lines of evidence suggest the feeding apparatus is well suited for this function. The slender shape reduces resistance during both lateral motion and jaw adduction when feeding underwater, and the increased snout length allows for a faster strike (*Pooley, 1989*; *Thorbjarnarson, 1990*; *McHenry et al., 2006*; *Pierce, Angielczyk & Rayfield, 2008*). Highly caniniform teeth can quickly puncture fast-moving, compliant prey, and their elongate shape may also lower their mechanical resistance (Fig. 7C). The longirostrine condition, defined as a snout that is both slender and elongate (*Brochu, 2001*), resulted in increased tooth positions; *Tomistoma schlegelii* had more teeth than any other crocodyloid, and *Gavialis gangeticus* has the most teeth out of all crocodylians sampled. This cranio-dental morphotype may be prey-size prohibitive though, as larger prey could damage the slender rostrum while struggling. Their elongate mandibular symphysis results in a mechanical disadvantage against the forces produced by shaking and twisting prey (*Walmsley et al., 2013*). The gracile nature of the dentition means a lower bending strength, making them more susceptible to breakage from larger and/or harder prey as well. On rare occasions, large individuals have been known to take land-based, vertebrate prey (*Thorbjarnarson, 1990*; *Selvaraj, 2012*). This is most likely because the overwhelming size of these crocodylians allows their feeding apparatus to withstand the forces exerted by said prey.

The slender-snouted taxa had some of the lowest size- and shape-heterodonty of modern crocodylians, which is reminiscent of several other clades of aquatic predators. They share certain traits with the anisodont plesiosauromorphs (*Sassoon, Foffa & Marek, 2015*; *Kear et al., 2017*). Although these crocodylians are not anisodont in the strict sense (they all have some shape heterodonty), both taxa have elongate mesial crowns transitioning to smaller distal ones. These taxa also reflect similarities with the 'homodont' condition apparent in odontocete whales (*Rommel, 1990*), where all the teeth in the arcade possess a similar, peg-like shape. This condition is believed to be ideal for catching and holding, but not processing, small aquatic prey (*MacLeod et al., 2007*), as most prey items consumed are under 10% of their body length (*MacLeod et al., 2006*). A convergent reduction in size- and shape-heterodonty within these independently aquatic groups may indicate a transition from a multi-functional dental arcade to one almost exclusively for prey capture. This morphological condition is best exemplified by *Gavialis gangeticus,* as it is almost entirely caniniform along its tooth row and eats primarily fish (*Groombridge, 1982*; Fig. 7C). Members of *Mecistops cataphractus* and *Tomistoma schlegelii*, although also primarily caniniform, still displayed the linear shape change typical of other crocodyloids.

These species may consume prey that require relatively more processing with their distal crowns, and there are numerous reports of them eating prey such as crustaceans, turtles, and immature primates (*Brazaitis, 1973*; *Groombridge, 1982*; *Galdikas & Yeager, 1984*; *Selvaraj, 2012*).

Tooth shape may indicate differences in feeding behavior and processing ability, even though overlap exists in prey selection. *Alligator mississippiensis* and *Crocodilus niloticus* both consume a wide variety of prey, including both large and small mammals, crustaceans, fish, water fowl, snakes, turtles, and conspecifics (*McIlhenny, 1976*; *Pooley & Gans, 1976*; *Groombridge, 1982*; *Delany & Abercrombie, 1986*; *Hutton, 1987*; *Shoop & Ruckdeschel, 1990*; *Rootes & Chabreck, 1993*; *Elsey, Trosclair & Linscombe, 2004*; *Wallace & Leslie, 2008*; *Gabrey, 2010*). A comparison of controlled feedings of each of these species showed *Alligator mississippiensis* to fracture and consume noticeably more bovine skeletal elements than *Crocodylus niloticus* (*Njau & Blumenschine, 2006*; *Drumheller & Brochu, 2014*; *Drumheller & Brochu, 2016*). Our *Alligator mississippiensis* specimens was generally more molariform than *Crocodylus niloticus*. These teeth would have greater bending strengths to resist breakage when processing hard material such as bone.

## Fossil taxa and the appropriateness of analogues

Certain fossil taxa were reminiscent of modern counterparts. We expected the fossil *Caiman crocodilus* to be similar to its congenerics, due to the fact that these specimens are closely related and all consume insects, crustaceans, and fish (*Brazaitis, 1973*; *Groombridge, 1982*; *Da Silveira & Magnusson, 1999*). Any differences in size and shape ranges appear to simply be a consequence of the former's incomplete arcades; no distal maxillary or dentary crowns were available (see **Limitations** below). *Alligator mississippiensis* specimens have similar shape regression statistics to our *Leidyosuchus canadensis* specimen, but, unlike members of *Alligator*, this specimen lacked enlarged distal teeth (Fig. 7D). This caused size heterodonty to differ noticeably, and may be indicative of a difference in the degree these taxa process hard materials (although no taphonomic evidence for this currently exists associated with *Leidyosuchus canadensis*). The two specimens of *Borealosuchus sternbergii* differed from one another in median shape as indicated by $y$-intercepts, which may due to an allometric increase in molariformy. The best analogue for this species may be a member of *Crocodylus* with similar slopes such as *Crocodylus palustris*, but more data are necessary to confirm this (Fig. 7D).

*Hamadasuchus rebouli* had similar relative size-heterodonty and relative maximum tooth size to the larger *Crocodylus porosus* specimen, which indicates it may have dealt with similar prey from a mechanical standpoint (Fig. 7E). The greatly enlarged mesial teeth would puncture vertebrate tissue with similar effectiveness (Fig. 7B). *Hamadasuchus rebouli* differed in that it had very large distal crowns, which, unlike *Osteolaemus tetraspis*, were laterally flattened (*Larsson & Sidor, 1999*). This suggests potential differences in prey processing. Peirosaurids are believed to be primarily terrestrial crocodyliforms (*Tavares et al., 2017*), and they most likely did not occupy the semi-aquatic, sit-and-wait predator niche dominated by modern crocodylians (*Larsson & Sues, 2007*). It may have used these

for either sheering soft tissue or breaking bone similar to modern mammalian carnassials, as rolling on land is not an effective means of dismemberment (*Fish et al., 2007*).

Several authors have stated that modern taxa do not have, or have secondarily lost, an extreme degree of molariformy commonly found in extinct representatives. 'Globidonty' describes the enlarged, highly molariform crowns in fossil taxa potentially used for durophagy (*Norell, Clark & Hutchison, 1994*; *Brochu, 1999*; *Brochu, 2001*; *Ősi & Barrett, 2011*). Species of *Brachychampsa* are textbook examples of a globidont taxa (*Case, 1925*; *Carpenter & Lindsey, 1980*; Fig. 7F), and our specimen is the only one in the sample with distal teeth so molariform their PC1 scores exceed 0.349. Although we agree with *Brochu (2001)* and *Brochu (2004)* that *Osteolaemus tetraspis* is not as extreme, the PC1 scores of its enlarged distal teeth are closer to our *Brachychampsa* sp. than any other taxon sampled (0.322–0.341). *Alligator prenasalis* and *"Crocodylus" affinis* distal crowns are similar to *Alligator sinensis* in shape, and also create a ridge of robust teeth (*Mook, 1932*). The mechanical capabilities of these particular crowns in modern taxa should be similar to the extinct, which suggests similar processing abilities in the distal regions of the skull. The similarities break down when the rest of the jaw is considered though. In addition to these highly molariform teeth, modern taxa also possess caniniform mesial teeth suggesting a division of labor along the tooth row. Contrarily, almost all teeth of members of *Alligator prenasalis, Brachychampsa* sp.*, and "Crocodylus" affinis* are on the molariform half of the shape-spectrum (Fig. 7F), making both their size- and shape-heterodonty rather low. These extinct taxa probably did not need to do as much puncturing of compliant substrate, which supports the argument that they may have foraged for mollusks and slow moving turtles (*Carpenter & Lindsey, 1980*; similar to *Salas-Gismondi et al., 2015*) rather than being ambush predators.

## Limitations and future work

A complete tooth row with all positions represented would be the most thorough way to assess heterodonty in any specimen. Nevertheless, complete tooth rows may not be available under many circumstances. Although we cannot propose a threshold for what number of teeth is 'enough' to accurately assess heterodonty, there are factors that influence some of these methods more than others. Foote's morphological disparity relies on, among other things, the grand mean and the sample size. Size-heterodonty may be underrepresented if, for example, an enlarged tooth is missing. This tooth would deviate greatly from the grand mean if present, so its exclusion would deflate size heterodonty. Due to their greater frequency, small teeth would typically be closer to the grand mean. Therefore, if a single small tooth was missing heterodonty would slightly inflate. It should also be noted that if the majority of small teeth were missing their absence would end up decreasing heterodonty, as the grand mean would approach the value of the remaining large teeth. Shape-heterodonty will not be as influenced by enlarged teeth missing, but more so by a lack of the mesial- or distal-most teeth. Missing many of the caniniform or molariform teeth will deflate shape-heterodonty, as the overall variance would be reduced. This would also influence regression statistics, as a lack of either extreme would raise or lower the slope and/or $y$-intercept.

 

Even though our data set included several specimens with tooth rows with less than 70% completeness, we feel our data is reliable enough for the morphological and functional conclusions we draw. The similarity between the heterodonty regressions representing all specimens, versus those with more complete specimens only, suggests that incompleteness did not influence our heterodonty values very much. This is most likely because the factors that would strongly influence size- and shape-heterodonty mentioned above were relatively rare in our data set. Few specimens sampled lacked enlarged teeth. The teeth that were missing were typically spread throughout the tooth row, and not localized to the mesial or distal extremes. There were several specimens whose heterodonty results were probably strongly influenced. The very low size-heterodonty in *Alligator prenasalis* may be a consequence of the loss of enlarged teeth, but the relatively small P4 crown suggests this value should be on the low-end regardless. The fossil *Caiman crocodilus* was also affected by missing teeth. It shared almost identical tooth morphology with modern congenerics at similar positions. Because the distal 30% of its teeth were missing, Foote's disparity of shape was reduced. This also affected the shape regression, as the lack of high PC1 scores on the distal end reduced the slope. Although both *Crocodylus siamensis* specimens had very high shape-heterodonty due to having both very caniniform and molariform crowns along their arcade, one specimen (AMNH 49231) yielded a very high shape disparity value. This was most likely partially influenced by missing teeth.

The two-block PLS test did not factor in heterodonty. Although this was not the goal of this particular analysis, it should be noted that the averaged tooth used for one block essentially eliminates variability of tooth shape along the arcade. Specimens with both low and high shape-heterodonty could produce very similar averages. This highlights the pitfalls of simply averaging all the teeth in the arcade together for the purposes of understanding dental morphology, and why it is avoided in our assessments of heterodonty here.

We did not consider all three dimensions. Living crocodylian teeth are often described as conical (*Edmund, 1969*) or conidont (*Hendrickx, Mateus & Araújo, 2015a*). Studies of bending strengths show variation between mesial-distal and labial-lingual axes (*Monfroy, 2017*), indicating that functional information may be drawn from the dimension not measured here. This is especially important concerning fossil taxa, as pronounced lateral compression is commonplace. *Hamadasuchus rebouli* distal teeth have been referred to as ziphodont (*Larsson & Sidor, 1999*), but, as this is defined by lateral flattening, our method did not register this character. Future studies should consider this third dimension at least qualitatively, in order to avoid conflating disparate tooth morphotypes such as these.

Principal component scores as shape measurements are very much dependent on the nature of the sample. Although they are very revealing concerning shape variability, they are not transferable between different data sets. A potential method for creating transferable shape metrics is using our PC1 axis as a guide to derive linear distance measures that would account for the serial shape homology seen in Crocodylia. Since PC1 essentially represents molariformy vs. caniniformy from the labial perspective, it could possibly be simplified into a comparison of linear distance measures such as maximum mesial-distal lengths and apical-basal heights. These metrics would be not only easy to collect, but also transferable between data sets.

These limitations aside, future researchers may apply aspects of this method to a host of non-mammalian taxa. As stated above the semilandmark analysis is very versatile, especially for single-cusped teeth. Foote's disparity can be applied to any measurable taxon with a complete enough tooth row, and is transferable between even disparate groups. Mesial-to-distal linear regressions may be directly applied to any crocodylian, and perhaps certain crocodyliform, specimens with an intact enough tooth row. This could also be applied to PC scores in other non-mammalian taxa, to determine what degree (if any) shape change is linear.

We limit our evaluation of interspecific differences, and make no attempt to analyze other factors such as ontogenetic changes (*Erickson, Lappin & Vliet, 2003*; *Erickson et al., 2004*; *Gignac & Erickson, 2014*; *Drumheller, Wilberg & Sadleir, 2016*). These variables may be investigated in the future using our method, as there is nothing to suggest that crocodylian individuals of most species and/or ages could not be be quantified in a similar manner. This method could be very useful in dealing with incomplete fossils. It is common for fossil crocodylian specimens to be lacking many, or even most, of their teeth. The linear nature of tooth shape can predict the shape of these missing teeth. A record of the ranges of slopes may be accumulated for fossil specimens with intact teeth. These slopes may then be used as a reference, and be applied to a fossil with the missing teeth. The preserved teeth can be plugged into the linear equation, and the shapes of missing teeth may be predicted with a high degree of certainty. This would result in a more complete representation of the extinct animal's anatomy, useful from the standpoint of both anatomical science and paleontological reconstruction.

Quantifying the teeth of Crocodylia will add rigor to future life history studies of the clade. First and foremost, values may be applied to the plethora of qualitative terms used by researchers (see **Introduction**). This would allow for stricter definitions of the terms when used in the future. As a quantifiable trait, both tooth shape in a single position and heterodonty as a whole may be incorporated into character matrices for phylogenetic analyses. Quantitative descriptors of dentition can describe a numerical range of morphology as opposed to cherry-picking an average tooth or single position. The teeth of fossil taxa can be compared statistically to modern taxa to determine the best analogue, and rigorous hypotheses about paleobehavior and paleoecology may be drawn. Crocodylia, both living and extinct, may be grouped into dental categories, allowing for species and specimens to be compared to one another (similar to snouts in *Brochu, 2001*). Frequency, size, and hardness of food items may be compared to these categories to determine if a link exists between dental morphotypes and dietary patterns (similar to *Aoki, 1989*). Crocodylians are used in both performance and actualistic taphonomy studies frequently (*Njau & Blumenschine, 2006*; *Erickson et al., 2012*; *Erickson et al., 2014*; *Drumheller & Brochu, 2014*; *Drumheller & Brochu, 2016*), and the output of these studies could be correlated with tooth dimensions. Tooth shape may also be compared to bite-force, death-rolling, bone-modification, and prey preferences and mechanical properties.

## CONCLUSION

Multiple measures of morphology have allowed us to describe heterodonty in a thorough manner across a number of both extinct and extant crocodylian specimens. Through a combination of Foote's morphological disparity and regression analysis along the tooth row, our data indicates that crocodylians are indeed heterodont with a number of dental morphotypes available spanning from extreme cases of caniniform to molariform. This variability may be functional in nature, and relate to the size, frequency, and compliance of certain prey in their typically generalist diets. The methods used here should be applied in the future to most crocodylian specimens, as well as other non-mammalian tetrapods, to investigate dental morphology in the context of a number of natural history related questions.

### Institutional abbreviations

**AMNH**   American Museum of Natural History, New York, NY
**ROM**   Royal Ontario Museum, Toronto, ON
**UCMP**   University of California Museum of Paleontology, Berkeley, CA

## ACKNOWLEDGEMENTS

The American Museum of Natural History Department of Herpetology, Royal Ontario Museum Vertebrate Paleontology, and University of California Museum of Paleontology curatorial staff allowed on-site access to dry skull specimens. We would like to particularly thank D Evans, P Holroyd, D Kizirian, C Raxworthy, K Seymour for their assistance.

### Funding

This work was supported by Daemen College. The University of Tennessee paid for the publication costs. The funders had no role in study design, data collection and analysis, decision to publish, or preparation of the manuscript.

### Grant Disclosures

The following grant information was disclosed by the authors:
Daemen College.
The University of Tennessee.

### Competing Interests

The authors declare there are no competing interests.

### Author Contributions

- Domenic C. D'Amore conceived and designed the experiments, performed the experiments, analyzed the data, contributed reagents/materials/analysis tools, prepared figures and/or tables, authored or reviewed drafts of the paper, approved the final draft.

- Megan Harmon conceived and designed the experiments, performed the experiments, analyzed the data, contributed reagents/materials/analysis tools, authored or reviewed drafts of the paper, approved the final draft.
- Stephanie K. Drumheller performed the experiments, contributed reagents/materials/-analysis tools, authored or reviewed drafts of the paper, approved the final draft.
- Jason J. Testin authored or reviewed drafts of the paper, approved the final draft.

## Data Availability

Dryad: doi: 10.5061/dryad.hk54vd3.

The raw data is also provided in the Supplemental Files.

## Supplemental Information

Supplemental information for this article can be found online at http://dx.doi.org/10.7717/peerj.6485#supplemental-information.

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
