# Peer review of "Quantitative heterodonty in Crocodylia: assessing size and shape across modern and extinct taxa"

_PeerJ, doi:10.7717/peerj.6485_

## Round 0.1 · original submission · Major Revisions

Dear authors,

I am sorry for the delay in this decision. I have accepted the decision of 'major revision' from the three reviewers.

Once again, thank you for submitting your manuscript to PeerJ and I look forward to receiving your revised submission.

Reviewer 1 ·

Basic reporting

1. The manuscript includes incomplete sentences, particularly early on in the main text. Please make sure that the authors proof-read the manuscript carefully before re-submission. I've listed minor errors in the "General comments" section but it's likely that I did not catch all minor errors.

2. Avoid the use of slash character “/” and instead, write out “and” or “or” (e.g., Ln 21, 39)

3. The Introduction section should be tightened with respect to narrative and logical flow. The section jumps back and forth discussing quantitative vs. qualitative; literature survey; and how previous quantitative work on non-mammalian teeth has been used. I suggest re-organizing the Introduction section so that these individual topics are organized into paragraphs and discrete chunks for efficiency and clarity of understanding.

4. The Introduction section should present justification and rationale for the study. For example, how is this study an improvement over previous studies? And why is geometric morphometric method being used?

5. The goals of the study are too vague and, in my opinion, not effective at providing a clear setup for the study. For example, what is meant by “multifaceted” (Ln 99), “developmental consistency” (Ln 100), and “individuals” (individual specimen or taxon (Ln 102)? These goals should be far more specific and give readers a clear impression of the hypotheses and topics covered in the study.

6. In my opinion, presenting the regression coefficients for skull against tooth shape (e.g., Ln 242; Fig 3) seems unnecessary since the relationship of PC1 values is not useful for further research (i.e., PC1 is not a very interpretable variable beyond this particular study).

7. The authors state that “size varied significantly between positions” (Ln 284). The word “significant” or “significantly” implies statistical significance. Was the variation across positions tested statistically? Similar statements are made in Ln 299 and possibly other places in the manuscript. I suggest revising these statements if statistical tests were not actually conducted.

8. Please provide an explanation as to why the Zahnreihe pattern appears “to be unrelated to the morphological variables investigated here” (Ln 359–360).

9. Much of the Discussion section seems too ‘adaptationist’ to me, although this is largely a personal stylistic issue. What would strengthen this section greatly is to construct a morphospace of tooth shape and see how they cluster according to species or ecological types. I feel this is essential for making the points authors make in the Discussion section. With a morphospace, for example, one can see to which extant taxon is a fossil specimen closest morphologically. This would help with the subsection on modern analogues for extinct crocodylians.

10. Somewhere in the Discussion section, there should be some discussion on variation in tooth count (e.g., Brown et al. 2015 J. Anat.) and its implication for this study.

Experimental design

11. It’s not clear to me how the tooth numbers were assigned based on the description given (Ln 122–124). From P4, how was P5 or M1 determined? Additional statement(s) clarifying the numbering system would be very helpful to fully understand this step of the study.

12. Is there a justification for the “40% measurable teeth” cutoff point for sampling the specimen (Ln 130–131)? If so, please mention this in the manuscript.

13. The statement “Specimens where the only teeth available biased towards the upper or lower size extremes were excluded” (Ln 131–132) is unclear to me. Please clarify.

14. I appreciate that juveniles were excluded from sampling, but how were “juveniles” specimens determined?

15. I am concerned about the use of two different cameras and (more importantly) camera lenses because parallax could incur significant artifacts to resulting shape data. The potential effect of parallax needs to be evaluated and reported in the paper to confirm the validity of the shape analysis in this study.

16. How was the number of semi-landmarks determined? Also, it was not entirely clear from the Materials and Methods section that the apical point is treated as a fixed (discrete) landmark.

17. Could the bilateral symmetry of the jaw and teeth be truly assumed (Ln 159–160)? Depending on the sample size of corresponding teeth, this should be tested and reported in the manuscript. Also, were shapes of corresponding left and right teeth averaged as well as centroid size?

18. There needs to be a specific description of how the orientation of the teeth and skull, as well as the camera, was standardized. This is a critical information for any morphometric analysis using photographs.

19. I wonder if centroid size is an appropriate metric for tooth size. Centroid size places more weight to (semi)landmarks at ends of structures due to the sum-of-SQUARES calculation. Therefore, a very narrow tooth can have exaggerated centroid size values. I would like to see the degree of correlation between, for example, tooth height and centroid size to confirm that use of centroid size is not introducing any artifacts to the analyses.

20. “All CS values were divided by it [head length] to normalize relative tooth size”. I would think that absolute size would be biologically meaningful. Please provide justification for normalizing the centroid size this way.

21. “Photographs of intact skulls of conspecifics were used instead [of incomplete fossil skulls] to derive head shape and length”. This statement immediately raised a red flag for me because the size and growth stage may be completely different between conspecific specimens. Please provide additional details and justification for this step.

22. Foote’s morphological disparity is a summary statistic of multivariate data. As such, I am not sure what the authors mean by “it [Foote’s MD] is not descriptive of multivariate measures such as shape” (Ln 201–202). Furthermore, the authors mention that to rectify this ‘issue,’ they use PCA to show shape variation in reduced dimensional space (primarily just PC1). This argument does not make sense to me unless they use the full dimensionality of original shape data.

23. Correlations between skull and average tooth shape should be performed with two-block partial least squares analysis that can analyze in full shape spaces.

24. Use of qualitative descriptions with highly subjective cut-off points (Ln 234–236) seems counter to the goals of this study. The need for associating words to numbers should be justified. Minimally, the actual values should be reported whenever these qualitative terms are used throughout the manuscript.

25. Given the undulatory pattern of tooth shape and size variation across tooth row, the use of linear regression seems inappropriate to me on a fundamental level, although it may model the overall shape changes adequately in practice based on what we see in Fig 5. Nevertheless, a critical problem is that non-significance of shape-to-position relationship does not mean lack of heterodonty because undulatory changes across tooth position (i.e. still heterodonty) can yield a non-significant linear model. The authors should provide a rationale for using linear regression analysis for investigating shape changes across tooth position.

Validity of the findings

26. There are many assumptions being made at all levels of the study, and thus, it is very difficult to assess the validity of the findings. From what I can see, the assumptions and potentially questionable practices in this study include, but not limited to: (1) mixing of captive and non-captive individuals; (2) suitability of semi-landmark sampling used in this study; (3) artifacts related to parallax from using two different camera lenses; (4) lack of detail on how photographing of specimens was standardized; (5) incompleteness of teeth across samples; (6) lack of clear definition of "juvenile" specimens; (7) assumption of perfect bilateral symmetry; (8) use of conspecifics to have size and shape data for incomplete specimens; (9) use of linear models to model shape across dental arcade; and (10) PC1 as adequate representation of variability across all tooth positions.

27. Authors state that shape and size heterodonty are decoupled (e.g., Ln 353), but to my knowledge, this was not explicitly tested in this study. This is easy to do with multivariate regression of shape onto centroid size.

28. Authors state that the study provides “a numerical range of morphology as opposed to cherry-picking an average tooth or single position” (Ln 589). While this is partially true, I would not consider analysis based on only PC1 of shape to be much useful for continuing the study. PC1 values are only particular to specific sample.

Additional comments

D'Amore and colleagues present a modern quantitative approach to investigate tooth size and shape variation in crocodilians.They employ a 2-D geometric morphometric approach and find results that support previous qualitative studies on the topic, as well as discovering statistically significant tooth shape differences among species and similar tooth shape variation in upper and lower jaws. The content of their study would be appropriate to be published in PeerJ and would benefit researchers working on crocodilian biology and dental and feeding adaptations. However, I found many serious issues and methodological details that need be reported and addressed. I have listed general and major comments above and also list more minor, specific comments below:

Ln 18, 98: insert “of” between “members” and “Crocodylia”
Ln 31: Change “gengeticus” to “gangiticus”
Ln 35: insert “are” between “shapes” and “typical”
Ln 35: Is the “former” referring to “Leidyosuchus and Borealosuchus” or “Alligator”?
Ln 36: “Hamadasiuchus rebouli similarities” should be “Dental similarities of Hamadasuchus rebouli”
Ln 39–41: Be more specific about the implications of this study.
Ln 62: Change “been” to included”.
Ln 63, 65: “Euclidean distances” should be changed to “linear distance measurements”. This would be more precise because Euclidean distance could be distances in multivariate space (e.g., shape), which is the basis for Foote’s disparity metric. Please revise other instances of “Euclidean distances” in the manuscript.
Ln 71–72: These descriptive words should be in quotations if they are direct quotes from the literature.
Ln 99: What is meant by “multifaceted”? Be specific.
Ln 121: One does not “atrophy” a structure. The structure becomes atrophied.
Ln 123: Add “tooth count” or “tooth assignment” after “standardize.
Ln 127: Change “resulting” to “resulted”.
Ln 129: “data” is plural, so “was” should be “were”.
Ln 139: Change “in” to “by”
Ln 157: Add “total” before “bending energy”
Ln 158: Generalized LS Procrustes superimposition is the over-arching procedure that would include the sliding semi-landmark step. Therefore, it is awkward (and inaccurate) reporting them separately as it is done here.
Ln 183: Ordination is a statistical approach, so should not be separated into “ordination and statistics”.
Ln 339: Change “morphotypes” to “morphotype”.
Ln 336: This sub-section label should go after the summary of main points (i.e., Ln 353).
Ln 394: Change “method” to “results”.
Ln 416: Would “considerably” be a more suitable word than “noticeably”?
Ln 458: In organismal biology, a structure is not “designed” for a function.
Ln 470: Add “of” after “reminiscent”.
Ln 544: The first letter of “Grand” should be lowercase. Also, I prefer to call it “pooled mean.”
Table 1: The variables should be explained thoroughly in the table caption.
Fig 1D: What does the deviation represent? For example, how many units along PC1. Have the deviations been magnified in any way?
Fig 3: Add confidence intervals for the regression lines.
Fig 4: Specify what the different colors indicate.

·

Basic reporting

This interesting piece of work aims to find a quantitative way to define and describe the hotly-debated concept of “heterodonty” in Crocodylia. The data collected mostly cover extant crocodilian species and some fossil taxa.
This novel method has the merit of breaking down the concept of heterodonty in quantifiable metrics describing both size and shape, and has immediate applications in interpreting the feeding style of extant and extinct animals. This topic has been long ignored and only vaguely dealt with in the past years, causing some confusion and the proliferation of different (qualitative) interpretations on the subject. With this regard, this work ought to be read by a broad audience including – and exceeding – zoologists, biologists, palaeontologist and ecologists.

I find this effort well thought and executed. The dataset is robust, the analyses appropriate and the few limitations of the method are openly acknowledged (although some clarifications are needed – see below). The importance of the finding is clearly stated in the background which is concise and requires only minor modifications. However, the results, and discussion are – in places – difficult to follow and should be clarified within the text.

Accordingly, my comments below largely concern the presentation of the finding, which I found difficult to follow in places. I suggested very limited changes and addition in the analyses for the authors, but I think that significant modifications are needed in the presentation of the results. The majority of my suggestions concern the organisation and presentation of the results and discussion, which I strongly recommend the authors consider to making this work clearer and more broadly accessible.

I overall enjoyed reading this manuscript and I’d recommend its publication upon satisfactory clarification of the detected issues.
I would be very happy to be contacted via email if the authors needed any clarification davide.foffa@ed.ac.uk

Davide Foffa

Experimental design

The research question is clear in the paper context. It is original, important and fits within the scope of the journal.This work is relevant and fills a gap of knowledge that has long been matter of debate. Rigorous tests have been performed using appropriate techniques.

However the methods, but also the results (and partially the discussion) are in place chaotically presented and need in my opinion re-organisation and tiding up. Although, the direction of the research is clear, its presentation is currently not as immediate.

The authors are openly transparent presenting their method in a comprehensive way, including a section listing limitations of the study.
Some clarifications are needed in order to ensure full replicability (see below and PDF).

Validity of the findings

Tindings of this work are meaningful, and based on an extensive dataset that could be the basis for future work. Their interpretation is also sound and supported by literature references. Further speculations are clearly identified as such.

The authors should add and appropriate Conclusion section because the paragraphs identified as such in the current version only describe merits, limitations and future directions of the method. This section is welcomed – and I think necessary (with some modifications), but clearer conclusions summarising the results of the study should also be added.

Additional comments

Numerous comments were left in the PDF attached. Besides those I have some concerns regarding some sections of the manuscript.
The phrasing of several paragraphs is awkward and difficult to follow, and there are few repetitions and organization issues within the text.
The use of terminology is often confusing. Terms are sometimes overlapping and are randomly introduced within the text (e.g. “procumbent teeth” vs “pseudocanines”). Each of these important nomenclatural terms should be unambiguously defined in an early section of the text (e.g. a nomenclature section in the Methods?), and consistently used through the manuscript to prevent confusion.
Figure and bibliographic referencing should be improved before publication.
The authors should for example decide whether they want to use abbreviations or complete generic names. Currently the same taxon is called interchangeably in 3 different ways (complete generic name, genus (spelled out) + species or genus (abbreviated) + species). Often in the same paragraph. If the authors will decide to adopt abbreviated generic names they should also make sure that they are unique (e.g. Caiman and Crocodylus should have different abbreviations).
Sentence repetitions and spelling should also be re-checked.

INTRODUCTION
I appreciate the effort that the authors made to summarise the vast background and descriptive terminology. This section provides a much needed summary of the state of the field and clearly identifies the rationale behind this study.
However, some of these terms are only mentioned in the introduction and there is no additional reference of them and how they relate to the current study. Others suddenly (and confusingly) reappear in the last section of the Discussion without being mentioned in the Methods or Results. It is thus unclear how the authors’ quantitative description of crocodilian dentitions reconciles with the past (largely qualitative) observations.
I suggested additions to qualitative descriptive labels to the figures, but I would also encourage the authors to add a table in which they compare their quantitative tooth types (high molariform to high caniniform) with the literature qualitative descriptors (e.g. do high-molariform correspond with globidonts)? Similarly with “pseudocanine”, “procumbent teeth” etc…

MATERIAL AND METHODS
This section has numerous clarity issues, that are fortunately easily solvable. The information here is presented in a fragmented way and early in the sections there are references to analyses that are explained long after.
For example at line 162, the authors explain the importance of rostrum shape, how they derive PC proxies for it and skill length, but they did not explain what these data are used for. That is explained later in the method section, leaving the reader unsure about the pertinence of such information until they have read the whole text.
The authors should re-think the logical presentation and succession of this whole section, systematically describing what data they used, for which analyses and why. I find that in the current version this information are scattered and difficult to follow.
I suggest that the addition of an “Institutional Abbreviation” (and perhaps a “Taxonomic Abbreviation Section”).
The morphometrics analyses done to the skulls are central to the discussion of this method and should absolutely be figured in a similar way as the tooth analyses have been (e.g. Fig. 1 -2 ) (see also PDF and figure section below).
The authors also use statistical jargon that should be avoided or at very least explained to facilitate the read to a non-specialist audience.
A list of PC scores and the % of variance they represent should also be provided for both teeth and rostra.
The taxonomic groups adopted for each specimen should be included at least in the supporting information.

RESULT
My biggest concern in this section is the PC-based description of the tooth shape (lines 233 to 236) and how they relate to the qualitative descriptors used in the literature – if they do. The authors should comment on that – ideally adding a comparative table (or figure) that shows the equivalence (if any) of their proposed terminology with the qualitative ones available in the literature.
Similarly, it should be stated how their PC results compared to “brevirostrine-longirostrine” transition. Ideally the use of these descriptive terms should be added in the relevant figures.
The organisation of some section is also confusing with an inconsistent succession of paragraphs that are sometimes organised by topic and sometimes by “extant vs fossil” (e.g. see comment on paragraph starting at line 272).
Rarely the sections in result and discussion start with a general statement on the finding, and often begin with in-detail description taxon per taxon.
Figure referencing should be improved and more frequent with references to specific sub-figures.

DISCUSSION and CONCLUSIONS
Also in this section the referencing should be improved and more frequent with references to specific sub-figures.
In places indicated in the PDF the authors should be more cautious in their explanations (e.g. “may be”or “likely is” instead of “is”).
The “Conclusion” is essentially a “limitation and future work” section, which should be incorporated in the discussion or left separated. Several points of the this section are not even mentioned (or should be expanded) in the main text (e.g. limitations, application to incomplete fossil specimens).
Nevertheless an appropriate “Conclusion” section summarising method, discussion, limitations and future work is currently missing and should be added
Refer to PDF for specifics.

REFERENCES
Reference formatting is inconsistent and inadequate, often missing key information such as volume, issue, and page numbers. Book references are incomplete. The entire section should be thoroughly checked.

FIGURES
The size of the figures should be clearly indicated. The format of sub-figures is inconsistent in the figure captions.
Figure 1. Add mesial, distal, labial, lingual labels. It would make a cleaner figure if the backgrounds of the skull and tooth figures were removed.
Figure 2. Half page width?
Specific comments on each figures are attached in the PDF

TABLES
The current format of the tables (landscape on a portrait page) should be re-evaluated and tables presented in a single page each

REFERENCES
References are inadequately formatted. Nearly one each two reference has some issues. Several entries are incomplete with inconsistent formatting, missing information (volume/issue and/or page numbers and/or book reference). Please, spend some time fixing these avoidable issues thoroughly. Double check again the alphabetical and chronological orders and the spelling of authors names and surnames.
There are also some in-text references in which the publication years do no match with the full reference in the bibliography.
I have pointed some of these mistakes out but there are likely many more.

Specific comments
Refers to PDF for specific comments.

·

Basic reporting

The English used in this manuscript is generally high quality and grammatically correct. Where possible I noted instances where sentences need to be modified for clarity and flow. The work conforms to high standards of courtesy and expression. There is sufficient background and citation of literature. There are a few instances where I suggest further exploration of the literature. For the most part section headings are consistent and professional. I noted one heading that was unlike the rest (line 237). Figures and tables are of a high quality.

Experimental design

The morphometric analysis is based on methods found in the published literature. The methods are sound. Data collection, specimen reporting, landmark schemes, and statistical analyses are well documented in the manuscript and should be clear to replicate if one wants to do so.

I do have a concern regarding that some specimens are sourced from unknown locations. I agree that few anatomical loci are located on crocodylian teeth and the landmark scheme does well in using loci that are homologous to all specimens in this analysis.

Extant specimens whose toothrows are incomplete should be avoided. I know of many specimens housed in other collections that could be used in place of the incomplete specimens used here. This is especially important as the authors admit that the incomplete toothrows may potentially inflate/deflate the morphological disparity of a species. Additionally, I would like to see larger sample sizes for each extant species, especially those species that are represented by a single individual in this analysis (Alligator mississippiensis, Gavialis gangeticus, Mecistops cataphractus).

I am slightly confused as to how snout shape was quantified - some clarification is needed in this section. I am also concerned about head length being used to calculate body length.

Validity of the findings

The findings presented here are well supported and valid. In the introduction the authors put forth four goals they would like to accomplish in their study but their second goal (line 100) is not well supported. They seek to document developmental consistencies found within the clade as a whole but did not include juveniles or a complete ontogenetic sequence for any of the species found in this study. This point should be elaborated upon or omitted.

Additional comments

Significant differences were found between the degree of heterodonty in extant crocodylians. However, captive specimens were not adequately controlled for by the authors. Captive rearing of crocodylians appears to affect the shape of the skull and teeth in these animals. This is problematic and affects the findings of this study. I would suggest that the authors only use animals that have records indicating that they were wild caught. Alternatively, if the animals with no information regarding their sourcing were proven to demonstrate little variance in skull and tooth shape relative to their wild-caught conspecifics I would be satisfied with their inclusion.

I disagree somewhat regarding the application of these methods in determining the shape of missing teeth in fossils. There may be too much variability present among toothrows of various species to accurately predict shapes. A good example of this would be a longirostrine taxon whose distal teeth are missing. It may be impossible to predict that the distals would be caniniform as found in Gavialis gangeticus or relatively more molariform as found in Mecistops cataphractus and Tomistoma schlegelii. Regardless of the complications, a relatively complete toothrow could be assigned with more certainty than a loose tooth, which unless diagnostic for a taxon would be impossible to assign with any measure of certainty.

Additionally, some terms need to be defined. Please add them to the nomenclature section under the Materials and Methods heading.



Comments by Section

Introduction
Line 55 - What is meant by size changes? I presume that you mean changes in the degree of heterodonty through ontogeny.
Line 100 – How were developmental consistencies explored? Juveniles were excluded and ontogeny is stated as beyond the scope of this study (lines 132-133).
Lines 101-102 – Distinguishing morphotypes between individuals is nebulous. Are these individuals of the same species or individuals from different species?
Line 123 – What is meant by procumbent? I would describe a tooth as procumbent if it projects labially however as used here it seems as if procumbent means a tooth that is proportionally larger than neighboring teeth – this term needs to be defined or a new term needs to be used as a replacement.

Specimens
Although correctly stated that captivity may have an affect on the morphology of the snout and teeth why was this not controlled for? It would be prudent to exclude all specimens that were not wild-caught or whose method of collection is unknown. I feel that including those specimens that were raised in captivity adversely affects some of the findings of this study, especially those that seek to aid in ecological predictions, dietary behaviors, and life history characteristics.
Lines 131-132 - Why were individuals whose only remaining teeth representing the largest and smallest size extremes excluded from this study? How were the extremes measured – relative to other individuals within the same species? How does their exclusion influence the findings of this study?
Lines 135-136 - It is understandable that incomplete toothrows were included for fossil taxa but why for extant taxa if incomplete toothrows inflate/deflate morphological disparity? Complete specimens could be found elsewhere.

Data Collection
Why were landmarks representing the occlusal surface excluded? Is it unnecessary in light of the information gained from the labial perspective? Additionally, two teeth that are similarly shaped from a labial perspective may have divergent occlusal surfaces. An example I can think of are two mesiodistally broad teeth, from a labial perspective, which may have flattened or blade-like occlusal surfaces respectively.
Lines 162-180 – There is some confusion about how snout shape was quantified, being explicit would help to avoid this confusion. Did you use pre-determined snout shape categories from Drumheller (2016) and Wilberg (2017) as stated on lines 165-167 or did you use the method described on lines 167-168? Should the latter be used to determine snout shape then it is inappropriate as the snout lies anterior to the orbits but the measurement extends to the posterolateral corner of the skull.
Line 174 – There are a number of considerations when using head length to measure body length. Ratios of cranial length to body length may differ between longirostrine, mesorostrine, and brevirostrine forms (Hutton, 1987; Fukuda et al., 2013). The ratio of head to body length may also change through ontogeny (Fukuda et al., 2013). Some have said that the femur is a better estimator of body length (Farlow et al., 2005).

Ordination and Statistics
What is the permutation test for? What is the null hypothesis? I assume that it is testing the association of Procrustes distance between the cranial and mandibular teeth.
Lines 198-199 – What is being normalized?
Lines 207-209 – By PCs do you mean PC scores? If so, explicitly state that you are comparing PC scores for the regression analysis.



Results

Shape Variability in the Sample
Why is the first PC the only one considered? Is it the only PC that accounts for greater than 5% of the variance as stated much earlier in the paper?
Line 229 – How is the crown elongate? Mesiodistally or basoapically?
Lines 233-236 – How were the corresponding values determined for the descriptors? Were the values assigned randomly, were there gaps present between the values in the data, etc.?
Line 237 – This heading is unlike the rest.

Foote’s Disparity and Morphotype Ranges
Line 258 – Gavialis gangeticus
Line 276 – Here and elsewhere properties are assigned to higher taxa. Species interact with their environments, higher taxa do not. Instead of saying “Leidyosuchus” has a property say that “species of Leidyosuchus have that property.”
Line 278 – Species of Crocodylus, see above.

Heterodonty Along the Toothrow
Line 289 – Replace has with have.
Line 290 – Insert ‘is’ between peak and relatively.
Lines 291-292 – Is this variability present in specimens of different species or within species?

Discussion

Developmental Trends in Crocodylian Heterodonty
Line 387 – Omit ‘the’ between by and captive.
Line 398 – Omit ‘a’ between with and specific.
Line 424-427 – It is possible that the proportion of hard prey items in a species diet has a more profound influence than the occasional predation of a hard-bodied prey item – this is alluded to on lines 433-434. Presumably species with a higher proportion of hard prey items in their diet should have more molariform distal teeth.
Line 439 – Define compliant in this sentence or elsewhere in the paper. This word comes up often and I am unsure what it means. It appears to mean that the prey item is subject to puncture.
Line 442 – What Brochu (1999) is referring to as pseudocanines are the enlarged third and fourth dentary teeth found in species of Diplocynodon. As used by Brochu this term has a limited definition.
Line 448 – The authors are making an assumption that all processing is accomplished by the distalmost teeth.
Line 466 – M. pterygoideus posterior is smaller in the long snouted forms (Endo et al., 2002) but M. adductor mandibulae externus profundus is generally larger in these forms (Iordansky, 1973; Holliday & Witmer, 2007).
Line 474 – Which groups? Gavialoidea and Crocodyloidea?

Fossil Taxa and the Appropriateness of Analogs
Line 499 – Species of Borealosuchus and species of Leidyosuchus. Assigning properties to higher taxa, see above.
Lines 525-526 – Species of Allognathosuchus and species of Brachychampsa.
Line 534 – Add ‘are’ between affinis and on.
Line 538 – Species of Allognathosuchus.
Lines 542-544 – Species of Allognathosuchus. Brochu (2004) found the genus to be a wastebasket taxon where many taxa were assigned based on characters that are plesiomorphic to the base of Alligatoroidea, the result of his study was to restrict the membership of the genus. Some of the dental variation found in this genus may be a product of historically including taxa that are not phylogenetically allied with the genus.

Conclusion
Lines 577-585 – I disagree somewhat regarding the application of these methods in determining the shape of missing teeth in fossils. There may be too much variability present among toothrows to accurately predict shapes. A good example of this would be a longirostrine taxon whose distal teeth are missing. It may be impossible to predict that the distals would be caniniform as found in Gavialis gangeticus or more molariform as found in Mecistops cataphractus and Tomistoma schlegelii. Regardless of the complications, a relatively complete toothrow could be assigned with more certainty than a loose tooth, which unless diagnostic for a taxon would be impossible to assign with any measure of certainty.



Tables and Figures

Table 1.
Crocodylus siamensis – host bone – mandible runs onto the second page, this looks awkward. It may be better to have this species on the next page.
There are two numbers (1 and 2) that are found at the bottom of the first table. What are they for?

Table 2.
There are two numbers (1 and 2) at the bottom of the first part of the table and one number (3) at the bottom of the second part of the table. Are they supposed to be there?
Close off the margins for all tables with lines. Some sections of the tables are open ended.

Figure 1.
The numbers representing the landmarks and semi-landmarks are too small to read.

Figure 2.
I am used to “Principal component one” being reported as the first principal component.
These are vector diagrams but not a principal component diagram. I would change the title of the figure.

Figure 3.
Which specimens do the vector diagrams represent? It is not clear.
The vector diagrams may look better as their own figure. As presented here it looks chaotic.

Figure 4.
Why are some of the fossil taxa not represented in the MD (size) or Centroid Size figure? I could not find an explanation in the manuscript.

Figure 5.
Gavialoidea not Gavialoidae in the figure caption.
In this figure CS is straightforward in that it is measuring size. However, the first principal component is presumably measuring the variability in shape. It is not obvious in the figure caption or the body of the manuscript describing the findings that go along with this figure.

Figure 6.
I would like to see all specimens in a lateral perspective. Some are from dorsal or ventral perspectives – consistency would make this figure look more appealing.
The length of the scale bars needs to be indicated in the figure caption. Make sure that the scale bars are of the same weight.

---

## Round 0.2 · Minor Revisions

Dear authors,

I have accepted the decision of 'minor revisions' from all three reviewers. However, given their extensive comments, and that reviewer 3 still has reservations regarding the captive raised specimens. Moreover reviewer 2 has raised concerns about the change in title (re: Crocodyliformes) as only one fossil datum point is included. Thus, I would consider there still to be 'moderate revisions' needing to be made.

I look forward to receiving your revised version.

Reviewer 1 ·

Basic reporting

If number is less than ten, it is customary to spell out the number (e.g., Ln 135: “eight” instead of 8).

Ln 240: It’s not clear if “Figure 1F” is showing the shape variation along a PLS axis or simply depicting what a vector diagram looks like. If it’s the latter, I don’t think it’s perfectly appropriate to reference this sub-figure.

Ln 246: As with the above comment, I’m not sure that referencing “Figure 1D” here is appropriate if it is just showing what a vector diagram looks like. Instead, the explanation of what the figure is depicting should be explained in the figure that uses these diagrams to show shape variation.

Ln 279–280: Reporting the range of PC scores is not biologically useful or relevant. It’s purely a mathematical construct.

Ln 287–288: Factors do not account for proportions of shape covariance. Is this the R-value of PLS? If so, this value accounts for the proportion of shape variation of dependent shape variables by the independent shape variables.

Ln 304–305: One cannot state disparity increased twice as fast as size because these variables are not comparable metrics. I would remove this statement and only state the overall trend (positive, negative).

Ln 305–306: I don’t see the point of reporting that “[a]lligatoroids occur on both sides of the regression [line]”. These types of observations are typically mentioned when certain groups are either above or below the regression line. Unless there is a good justification for reporting this, then I would remove this sentence.

Ln 315–316: “Because of this poor correlation, we do not consider residuals as very biologically meaningful.” The biological meaning of residuals is not dependent on the degree of correlation. For example, one may choose to analyze residuals from allometric trend if they wish to analyze size-independent data regardless of the strength of the allometric signal.

Ln 473–480: In this section of the Discussion section, I would discuss the possibility of tooth pressure being a more important metric for functional capabilities than bite force (Erickson et al. 2012 PLoS ONE).

Experimental design

Ln 198: How was the coordinates mirrored? Which landmarks were used to define the median plane?

Ln 265–266: The comparison of slope based on regression of PC scores of shape on proportional position along the dental arcade is not sound. What is the biological meaning of slope? Why not use the full shape data to conduct a MANOVA? This is not difficult to do with available programs and computational tools and it is a far more preferred method when analyzing high-dimensional morphometric data. This way, one can also readily and more visualize the shape changes occurring along the tooth row than inelegantly translating PC1 scores back to qualitative descriptions of tooth shape, which also counters the aim of the study. That being said, results from PC1 can be arguably kept since it shows the degree to which tooth shape changes along the dental arcade on a plot. However, results from MANOVA should also be reported.

Ln 262: I don’t see how subtracting 0.5 from the proportional position along the dental arcade is necessary because this would not impact the statistical significance and slope coefficient of the regression analysis. Later, the authors claim that the “y-intercept would represent shape value for the median position, as the intercept is located half-way along the tooth row”. However, I do not see the biological significance of the intercept, especially when PC scores are used where linear combinations of shape data are defined purely mathematically (i.e., by variance).

As I commented on the previous version of the manuscript, I still do not think that performing a **linear** regression on centroid size and tooth position is sound when one knows that the size undulates across the tooth row. Although the authors inserted a caveat with this analysis (Ln 451–452), they misinterpret the results stating that “size varied significantly between positions” (Ln 337–338), but this is not what they test. They are testing whether size has a linear trend with tooth position.

Ln 399: “Regression analysis of shape is appropriate based on significance and high r2 values.” This reasoning is circular. R-square values do not dictate whether the use of regression analysis is valid. A theoretically and practically valid use of regression analysis can still yield low R-squared values.

Validity of the findings

Ln 272–275: R-squared of 0.09 for allometric signal in shape data is within a range of typical values expected in biological systems. Therefore, I would not consider the allometric signal to be “very weak […] overall.” This is because shape data comprise many variables, so the capacity of an explanatory variable to account for proportional variation diminishes.

Additional comments

The revised manuscript is considerably improved from the previous version. I appreciate the authors taking the time and effort to revise and refine their manuscript based on my previous comments. Below is a list of my more minor comments:

Ln 44: What is precisely meant by “moving target”? Please clarify in text.

Ln 74–76: Grammatically, each of these adjectives should be in quotations.

Ln 96–97: I don’t think the hyphen after “size” and “shape” are needed.

Ln 99: “geometric morphometrics” is the name of a field, not method. Change to “geometric morphometric methods/techniques/approach”.

Ln 112–116: Although it is fine to leave as is, these dental terms are firmly established in anatomical literature so I do not think the explanation is necessarily. Similarly, I don’t think Fig. 1B is necessary.

Ln 117: I would remove “independent of shape” because shape is not independent of size. These are strongly coupled aspects of morphology (i.e., allometry).

Ln 135: commas not necessary before and after “and 8 extinct”.

Ln 176: It is somewhat odd to mention “calculated centroid size” after “superimposition on the data” because the former is done before alignment. I would change the order of steps so that centroid size calculation is mentioned before superimposision. Also, the sliding of the semilandmarks is performed during the alignment procedure, so I would combine the two stating GLSP was done with sliding semilandmarks minimizing total bending energy.

Ln 260: Values between 0 and 1 are technically not a “percentage”. I would replace the word with “proportion”.

Ln 280: I would describe “negative-most condition” as “shape changes towards negative PC1 values include…”

Ln 317: Remove “relatively”.

·

Basic reporting

As the previous version the exposition is professinall and is appropriate, the literature is sufficiently cited (some comments on the pdf on potential additional references to be included). Overall the authors have done a very fine job and have been clarified the flow and contents of the text in numerous places. Intro, context and background have also benefited from substantial rewriting and simplification of the manuscript.
There are few issues with the figure format and in-text reference (see General Comments) and all data are supplied.

Experimental design

The design and execution of the study has also been simplified and it is considerably more focused now. The methods are clear, and should only be clarified in minor details (see PDF) but none of the required modifications affects the replicability of the experiment.
My major concern (but it is a small one), is that the authors should be more consistent in explaining the limitations of adopting incomplete specimens through the text. This is done very well in the final section of the manuscript but references to it should be present in the method section too.

The results and discussion are the parts that were mostly improved. The results are better linked with the methods and discussion. Furthermore the discussion has been largely expanded to include implication with the right amount of speculation (when needed). These sections are balanced and honest. The authors are very open on the merit and limitations of their method (perhaps understating in places the future possibilities of using it with in clades and associated with direct data on diets/ecology – particularly if the method will be expanded on fossil taxa).

Validity of the findings

No further comment from the previous version.
In the first and second submission this study fills in a gap in our knowledge and it does so in a robust, honest and meaningful way.
Once published this work will be surely applied to numerous other case studies.

Additional comments

General Comments
I am pleased to notice that many changes suggested by the reviewers have been implemented. I think the manuscript is clearer now and has a more specific and better-defined focus. I also agree with the authors that the new design of the study make some of the comments redundant of inapplicable, and I am pleased to notice that the authors spent time to deal with some of them anyway.

In my opinion there are still few problems that still persist, as minor issues, but I think they would be an easy fix. Nevertheless, I would like to see these issues addressed and I think that some of the authors’ choices should be better explained in the text.

I have left a pdf with some specific comments, the main of which are summarised here below.

a) I think that the title is currently misleading: the dataset is almost entirely made of crown crocodylians with only 1 crocodyliform (non crocodylian) specimen. In other words I don't think that the inclusion of a single non-crown crocodlylian justifies the change of the title from Crocodylia to Crocodylifomes. It probably is the case, but this dataset is, in my opinion insufficient to live up the title claim.
I would either change it back to "Quantitative heterodonty in Crocodylia: ..." or if the authors want to maintain the title as it is, they should increase their sample size by adding several more non-crown crocodylian specimens - with a (possibly) even distribution in the clade (e.g. several lineages represented). This may not be possible - depending on the fossil record.

Accordingly several citations of crocodyloforms should be double-checked.

This change may stem from a previous comment of mine in which I asked to explain the inclusion of Hamadasuchus rebouli in a crocodyilian-base study. A simple explanation of the fact is enough: perhaps Hamadasuchus is interesting because it is thought to have strong heterodonty and the authors wanted to check this in their study?

Related to this there is the sample description issue. The numbers of species, specimens, crown-crocodylian and non-crown crocodilian crocodyliforms is unclear and confusingly stated in different part of the manuscript (see pdf comments), and should be addressed.

b) The text would greatly benefit from many more in-text figure references that are very scarce at the moment. Ideally they would include sub figure references (e.g. Fig. 3A) as it is done in some parts of the manuscript, but not consistently.

c) There are weird grey/white halos and lines and grey blocks around all the figures – most likely left from formatting the images for publication. I imagine they may have passed unnoticed because of screen settings, but they are there and should be deleted. Please address and make the figure backgrounds a uniform solid white.

Further minor comments (mostly related to formatting and rephrasing) are included in the PDF.

Upon the fulfillment of these modification I will be looking forward to seeing this very useful manuscript published and I won’t need to review this manuscript again.

·

Basic reporting

The English used in this manuscript is for the most part grammatically correct. Below I noted instances where sentences need to be modified for clarity and flow. Background information is sufficient and literature is thoroughly cited, I noted a few instances where I think additional citations are needed. Section headings are consistent and professional. Figures and tables are of a high quality but there are a few changes that need to be made (see additional comments for each section below).

Experimental design

The morphometric analysis is based on widely used methods, there are a number of instances in which terms are being misused – I made note of them below. Data collection, specimen reporting, landmark schemes, and statistical analyses are well documented in the manuscript and should be replicable.

I am still concerned about some specimens being sourced from unknown locations as captive specimens may have skull and tooth morphologies that differ from wild-caught conspecifics. It was appropriate for the authors to address this issue in the second draft.
The results may be stronger with the exclusion of some of the very incomplete specimens. I understand that the fossil taxa are out of the author’s control but some of the specimens (Caiman crocodius UCMP 42844, Crocodylus palustris AMNH 75707, Crocodylus siamensis AMNH 49231) were missing more than half of the teeth of the upper jaws. What effect on the results does excluding these specimens have? Does keeping them in improve the results? An interesting example of the effect of incomplete specimens is found in Caiman crocodilus. Comparing the more incomplete fossil specimen of this taxon to the more complete modern specimens the author says “Any differences in size and shape ranges appear to simply be a consequence of the former’s incomplete arcades; no distal maxillary or any dentary crowns were available.” It would appear that incomplete specimens may negatively impact the findings of this study. I am glad that this shortcoming was addressed in the “Limitations and Future Work” section of the discussion.

As stated in my first review using head length as a proxy for body length is suspect. There are a number of considerations when using head length to measure body length. Ratios of cranial length to body length may differ between longirostrine, mesorostrine, and brevirostrine forms (Hutton, 1987; Fukuda et al., 2013). The ratio of head to body length may also change through ontogeny (Fukuda et al., 2013). Some have said that the femur is a better estimator of body length (Farlow et al., 2005). Comparing heterodony with head length is sufficient in my opinion and tells and interesting story without having to make leaps of faith regarding the determination of body length from cranial specimens. However, the authors addressed this shortcoming within the body of the manuscript.

Validity of the findings

Other than some concerns with experimental design, the findings presented here are well supported. The limited the scope of the latest draft made for a better product.

Additional comments

Captive rearing of crocodylians has a pronounced effect on the shape of the skull and teeth in these animals and should be obvious to a trained observer. From those specimens in figures 1 and 7 it appears that none of the figured specimens were sourced from captive populations. Should captive reared specimens be included they will negatively affect the findings of this study. If the specimens with no information regarding their sourcing were proven to demonstrate little variance in skull and tooth shape relative to their wild-caught conspecifics (of similar size) their inclusion is justified. I would like to see the datapoints in the various figures labeled (fig. 3 for example). This would make for a more informative figure and would help illustrate any specimens that may have aberrant values relative to their conspecifics.

The addition of a graph showing morphospace occupation would be a welcome addition to figure 2. This figure should show the taxa plotted in morphospace along the first and second principal components. Tooth morphology at the positive and negative extremes along the first PC should be placed at their respective ends of the axis. Doing so would make this easier figure more informative and easier to visualize. Plotting of morphospace, with minimally PC1 compared to PC2, is pretty standard in these types of studies.

Throughout the paper genus names are used in place of a specific species. For example Borealosuchus in place of Borealosuchus sternbergii. Unless one is referring to all taxa within the genus the binomial name should be used.

Below is commentary for each section of the paper with the lines in which the revision needs to be made.

Abstract

21-22 Change Principal Components Analysis to Principal component analysis
29-30 and 35-36 If slender snouted taxa have low shape heterodonty shouldn’t regression be adequate to predict the shape of their teeth?

Introduction

47 …issue has not been fully resolved.
59 Also (Ősi, 2014)

Nomenclature

113-114 Central premaxilla – I know what is meant here…where the premaxillae meet at the midline. However obvious to us, it could be misconstrued. I would reword this term.
116-117 Enlarged teeth are determined relative to other teeth in a given section… how are sections of teeth within the arcade being defined? Whether this was defined or not I missed it.
127-129 I understand that some taxa may lose a premaxillary tooth as they grow but for consistency it may be best to omit those specimens possessing more than 5 premaxillary teeth.
138-139 Postcrania, especially the coossification of the neural arches to the neurocentra may be used to determine maturity (if associated postcrania are not included with the specimens I understand). Diet changes over ontogeny and evidence suggests dental morphology does as well (Gignac and Erickson, 2014).

Statistical Approaches

216-217 The confirmation of the null hypothesis should be reported in the results section.
222 Report this in the results section.
226 Variable i – is this being compared within species or between species?
231-233 Allometry needs to be measured over ontogeny, no ontogenetic sequence used here.
236-238 This averaging technique could potentially produce an average shape that has no biological basis, in that the average shape may not represent any of the teeth found in the specimen. What effect does this have on the study?
242 Axis instead of axes?
244 Principal component analysis
248 Why is CS being normalized against head length? This is not clear to me.
255-256 Why was 7 chosen instead of another value?

Results

304 Possessed instead of possessing
305 Which clade is being referred to?
305-306 “Alligatoroids occur on both sides of the regression (line)” – as do crocodyloids.
308 Which species did this individual belong to? Was this significant or an aberrant specimen belonging to a species with multiple individuals in this analysis? I would like to see the data points numbered (or an alternative way to keep track of which specimen is represented by each data point) so one may reference the specimens in this study.
312 “greater” means larger?
314 “correlated significantly” Did you perform a hypothesis test to determine the significance of the correlation coefficient?
316 Omit ‘very’
320 Which specimen was the most shape heterodont? The one with most of the teeth intact or the one with more than half of the upper teeth missing?
328 Use Borealosuchus sternbergii instead of Borealosuchus (only one species used in this analysis)
333-335 Revise this sentence – “The vast majority of the modern taxa tooth rows had over 60% their tooth positions represented by measurable teeth (Supplemental Information Table S3).”
358 A PC score is reported as cartesian coordinates and must have two numbers. In your case the first would be the value along the first PC axis and the second would be the value along the second PC axis. If you want to report the value along one PC you cannot call it a PC score. Call it a value along a PC axis instead.
361 Revise this sentence for clarity.
365 Report this in cartesian coordinates if you want to use scores - or say that the species had certain values along the first PC.
366-367 Revise this sentence - Both taxa showed a similar progression from caniniformy to molariformy as the alligatoroids and crocodyloids.
377 “had the y-intercepts” - omit ‘the’

Discussion

390-391 I think referring to head length instead of body size is better practice. Head length to body length ratios are variable for the species.
444 Using the word interlock has a specific meaning to most croc workers, we think of the interfingering dentition found in crocodyloids. I would use another word to describe this morphological pattern.
540 These species instead of caimanine – do not assign properties to higher taxa.
557-558 Tomistoma schlegelii and Gavialis gangeticus, only looking at one species in each genus.
576-579 It seems as if an argument is being made for Gavialis (presumably Gavialis gangeticus) undergoing a reduction in size and shape heterodonty over its phylogenetic history. From my observations of its close and distant relatives this is not the case.
604 Leidyosuchus canadensis
610 Which species of Crocodylus?
636 Any instead of ‘another’

Figures

Figure 1 From my observations of Alligator and Caiman the fourth maxillary tooth is often the largest. In Brachychampsa and Crocodylus the fifth is the largest. You say as much on lines 340-344. Indicating that you are referring to the specimen in the figure and not to the sample as a whole would get rid of the confusion.
Figure 2 I would like to see this figure showing the taxa plotted in morphospace along the first and second principal components. Placing the tooth shapes representing the tooth morphology at the positive and negative extremes along the first PC (at their respective ends of the axis) would make this easier to visualize.
Figure 3B I would like to see which taxa each of the points belong to. Being able to visualize the taxa that the datapoints belong to would be informative for me. It’s possible that you could number the points and indicate which taxa they belong to – in this case little would have to be done to the figure to alter it.

Supplemental

It is suggested that the second PC describes the amount of “lean” with positive values indicating a tooth that leans mesially and negative values describing a tooth leaning distally. The second PC should be plotted, along with the first PC, in figure 2. Plotting of morphospace, with minimally PC1 compared to PC2, is pretty standard in these types of studies.
I’m assuming that PCs 25-114 represent a proportion of the variance in the data (even though reported as 0.00). Add more digits after the decimal point and report their values, no matter how small, or omit them from the supplemental data altogether.
Table S3 Change ration to ratio

---

## Round 0.3 · Minor Revisions

Dear authors,

Reviewer one has recommended ‘minor revisions’ for some comments that should be easy to fix.

I look forward to receiving your revised manuscript.

Reviewer 1 ·

Basic reporting

Basic reporting generally meets expectations with some minor issues which I list in "general comments".

Experimental design

Experimental design meets expectations.

Validity of the findings

The findings are valid.

Additional comments

The manuscript has been improved compared to the previous version. I appreciate the time the authors have taken to revise the manuscript. I only have minor comments that I would like the authors to address before accepting the manuscript for publication:

Basic Reporting

Ln 22: principal components analysis itself does not “visualize shape variance.” The statement should reflect the fact that principle components analysis is used to extract PC axes and scores that are then used to construct a morphospace that visualizes shape variance.

Ln 231–231: “The null hypothesis of bilateral symmetry was confirmed.” A subtle yet important distinction is to state that the test failed to reject the null hypothesis, as opposed to “confirming” the null hypothesis.

Ln 259: It is not clear what “shape coefficients” are referring to (e.g., full shape data or PC scores).

Ln 321: Insert “partial least squares” between “two-block” and “test”.

Ln 339–340: My comment on a previous version of the manuscript questioned the use of reporting the absolute value of the slope coefficient when regressing morphological disparity against skull length. The slope of “two” does not mean anything biologically because the disparity values for tooth size and shape are not comparable. As before, I recommend reporting whether the trend is positive or negative.

Ln 412: Instead of “strongly linear”, I suggest writing “shape heterodonty was found to strongly correlate with tooth position.”

Ln 439: “Discrete” is not an appropriate term here. “Continuous and quantitative” may be more precise adjectives to use in this case.

Ln 443–444: Measuring Procrustes variance (equivalent to Foote’s disparity) is not a less traditional approach in GM. In fact, it’s a core aspect of geometric morphometric analysis. I suggest removing the statement suggesting that the use of Foote’s morphologicaly disparity is a derived or innovative GM technique.

Ln 454: Replace “resultant” with “resulting”.

Ln 463: “Data” is plural, so the verb should reflect this: “data suggest”.

Ln 480–487: I recommend that the discussion on the potential effects of rearing conditions on the results should be discussed in the main text (i.e., not restricted to supplementary information). I believe there is useful information that is worth including in the main text.

Ln 605–607: The interpretation of the particular trend in tooth size in caimanines as an adaptation for “more mobile and/or compliant prey” is not clear to me. Please elaborate.

Ln 679: More data “are”, not “is”.

Ln 717: Remove “so” after “more”.

Ln 720–722: Not sampling small teeth would also have the potential to decrease size-heterodonty if the smaller teeth in general are not being collected. As such, the statement that not sampling small teeth would lead to less deviation from grand mean is inaccurate. Please revise the sentence.

Ln 746: Insert “partial least squares” between “two-block” and “test”.

·

Basic reporting

Nothing new to add.

Experimental design

Nothing new to add.

Validity of the findings

Nothing new to add.

Additional comments

Nothing new to add.

---

## Round 0.4 · accepted · Accept

Dear authors,

Many thanks for your revised manuscript. After reading it, I have accepted it for publication in PeerJ.

Once again, thank you for submitting your manuscript to PeerJ and I hope you will use us again as your publication venue.

If we need to clarify any details required to move the manuscript forward, then our production staff will get in touch with you. Otherwise, a proof will be forthcoming shortly for your review.

Congratulations and thank you for your submission.

#